

# Validation of GEMS tropospheric NO$_2$ columns and their diurnal variation with ground-based DOAS measurements

Kezia Lange[1], Andreas Richter[1], Tim Bösch[1], Bianca Zilker[1], Miriam Latsch[1], Lisa K. Behrens[1], Chisom M. Okafor[1], Hartmut Bösch[1], John P. Burrows[1], Alexis Merlaud[2], Gaia Pinardi[2], Caroline Fayt[2], Martina M. Friedrich[2], Ermioni Dimitropoulou[2], Michel Van Roozendael[2], Steffen Ziegler[3], Simona Ripperger-Lukosiunaite[3], Leon Kuhn[3], Bianca Lauster[3], Thomas Wagner[3], Hyunkee Hong[4], Donghee Kim[4], Lim-Seok Chang[4], Kangho Bae[5], Chang-Keun Song[5], and Hanlim Lee[6]

[1]Institute of Environmental Physics, University of Bremen, Bremen, Germany
[2]Royal Belgian Institute for Space Aeronomy, Brussels, Belgium
[3]Max Planck Institute for Chemistry, Mainz, Germany
[4]Environmental Satellite Center, National Institute of Environmental Research, Incheon, Republic of Korea
[5]Department of Urban and Environmental Engineering, Ulsan National Institute of Science and Technology, Ulsan, Republic of Korea
[6]Division of Earth Environmental System Science, Major of Spatial Information Engineering, Pukyong National University, Busan, Republic of Korea

**Correspondence:** Kezia Lange (klange@iup.physik.uni-bremen.de)

**Abstract.** Instruments for air quality observations on geostationary satellites provide multiple observations per day and allow for the analysis of the diurnal variation of important air pollutants such as nitrogen dioxide (NO$_2$) over large areas. The South Korean instrument GEMS on the GK2B satellite was launched in February 2020 and is the first instrument in geostationary orbit that delivers hourly daytime observations of NO$_2$. The measurements with a spatial resolution of $3.5\,\mathrm{km} \times 8\,\mathrm{km}$ cover a

large part of Asia.

This study compares one year of tropospheric NO$_2$ vertical column density (VCD) observations of the operational GEMS L2 product, the scientific GEMS IUP-UB product, the operational TROPOMI product, and ground-based DOAS measurements in South Korea. The GEMS L2 tropospheric NO$_2$ VCDs overestimate the VCDs retrieved from the ground-based observations with a median relative difference of +64 % and a correlation coefficient of 0.75. The median relative difference is -1 % for

the GEMS IUP-UB product and -14 % for the TROPOMI product. The evaluation of the GEMS IUP-UB product and the operational TROPOMI product with ground-based measurements is in good agreement with correlation coefficients of 0.82 and 0.88. The scatter in the GEMS products can be reduced when observations are limited to the TROPOMI overpass time.

The observed diurnal variations of the tropospheric NO$_2$ VCDs show a maximum of NO$_2$ during the late morning for urban sites, whereas rural sites show weak or almost no diurnal changes. Investigations of the seasonal diurnal variability show with

a minimum in the observed tropospheric NO$_2$ VCDs around noon the importance of chemical loss of NO$_2$ in summer. Most variability is seen in spring and autumn, which dominate the average annual diurnal cycle.

Observations under low wind conditions show strong enhancements of NO$_2$ over the day, especially at polluted sites during winter. This indicates that under calm conditions, dilution and the less effective chemical loss in winter do not balance the





accumulating emissions. The impact of transport processes is illustrated by the diurnal variability at a rural site following mean

wind patterns for specific seasons and observation times.

Analyzing the weekday-weekend effect, good agreement was found between the different products. However, the GEMS L2 product while agreeing with the other data sets during weekdays shows significantly less reduction on weekends.

Our investigations show that the observed diurnal evolution of NO$_2$ varies significantly at the different measurement sites, with good agreement between the GEMS IUP-UB and ground-based observations. The diurnal variability of tropospheric NO$_2$

VCDs depends on chemistry, emissions, and transport into and out of the measurement region. To interpret the sources and sinks of NO$_2$ requires that all of these factors are considered.

## 1   Introduction

Nitrogen oxides, in particular nitrogen monoxide (NO) and nitrogen dioxide (NO$_2$), collectively referred to as NO$_x$, are among the most important air pollutants and strongly impact tropospheric chemistry. NO$_x$ is emitted into the atmosphere by natural

sources such as lightning and soil microbial processes, but the primary source is anthropogenic activities. Anthropogenic emissions are caused by fossil fuel combustion mainly for transportation, the industry and energy sector, and residential heating (Seinfeld and Pandis, 2006; Wallace and Hobbs, 2006). High concentrations of NO$_x$ are a health hazard, which gets especially relevant as most anthropogenic sources are concentrated in urban areas with high population densities Faustini et al. (2014).

Tropospheric NO$_x$ is mainly emitted as NO, which is rapidly converted to NO$_2$ by the reaction with tropospheric ozone (O$_3$).

Due to their short atmospheric lifetimes, on the order of a few hours in the boundary layer during daytime (Beirle et al., 2011), the heterogeneous distribution of sources and variations of meteorological conditions, tropospheric NO$_2$ shows high spatial and temporal variability. Monitoring and understanding this variability is necessary to better understand the contributions of emissions, tropospheric chemistry, and transport effects, especially in urban areas with large and heterogeneous NO$_x$ sources combined with high population densities.

To resolve this spatial and temporal variation of tropospheric NO$_2$, measurements with good spatial and temporal resolution are needed. NO$_2$ can be remotely observed using the DOAS (differential optical absorption spectroscopy) technique (Platt and Perner, 1980). DOAS measurements of NO$_2$ have been performed from different platforms, including ground-based stations, moving platforms such as cars, ships, or aircraft, and environmental satellites, with advantages and disadvantages regarding spatial and temporal resolution.

Stationary ground-based instruments such as multi axis DOAS (MAX-DOAS, (see e.g., Hönninger et al., 2004; Wittrock et al., 2004; Herman et al., 2009)) can provide several observations of NO$_2$ column densities per hour but are limited to their location. These data sets are commonly continuous and are valuable for validation of satellite observations, among other applications (e.g., Pinardi et al., 2020; Verhoelst et al., 2021).

Mobile car DOAS measurements enable the observation of spatial variability in addition to its temporal evolution and are an

additional valuable source for satellite validation (e.g., Wagner et al., 2010). They fill a gap between stationary ground-based and satellite observations by mapping the variability within satellite pixels and quantifying errors for satellite and stationary





ground-based comparisons.

The advantage of measurements from environmental satellites in polar sun-synchronous low earth orbit (LEO) is that they can provide global coverage. The spatial resolution of satellite observations making use of the DOAS method has increased

strongly since the first mission with a ground footprint of $320\,km \times 40\,km$ for the Global Ozone Monitoring Experiment (GOME) in 1995 (Burrows et al., 1999) to the recent TROPOspheric Monitoring Instrument (TROPOMI) with a spatial resolution of $5.5\,km \times 3.5\,km$ (Veefkind et al., 2012). This offers the possibility to deconvolve sources of $NO_x$ such as individual power plants and to quantify their emissions (Beirle et al., 2019a). Satellite measurements also enable the seasonal variations of $NO_2$ to be observed globally. This has been done, for example, using SCIAMACHY (Bovensmann et al., 1999) observations

to disentangle the sources of $NO_x$ (van der A et al., 2008) or using TROPOMI observations to analyze the seasonality of $NO_x$ emissions and lifetimes (Lorente et al., 2019; Lange et al., 2022).

However, instruments in low earth orbits usually provide only one measurement per day and per location. Combining observations from several satellites with different overpass times provides some additional information on the diurnal variation of $NO_2$. Several studies have applied this method, based on the morning overpasses of the SCIAMACHY or GOME-2 (Munro

et al., 2006) instrument and the early afternoon observation of the Ozone Monitoring Instrument (OMI, Levelt et al. (2006)) (see e.g., Boersma et al., 2008, 2009; Penn and Holloway, 2020). Boersma et al. (2008) used SCIAMACHY and OMI data to estimate the diurnal variability of $NO_2$. Over urban regions, they found up to $40\,\%$ reduced $NO_2$ columns in the OMI afternoon overpass compared to the SCIAMACHY morning overpass. They explained this by photochemical loss, dampened by the diurnal cycle of anthropogenic emissions. Over biomass burning regions, they detected an increase from the morning to the

afternoon overpass, which is consistent with fire counts from the geostationary satellites. Analyzing the differences between SCIAMACHY and OMI tropospheric $NO_2$ columns from Israeli cities, Boersma et al. (2009) found again $40\,\%$ reduction for $NO_2$ columns in the afternoon compared to the morning overpass during summer, and nearly no differences in winter with only slightly higher $NO_2$ in the afternoon. Penn and Holloway (2020) found around 1.5–2 times higher $NO_2$ columns for the morning compared to the afternoon overpass for large urban areas in the US using GOME-2 and OMI observations.

To analyze the diurnal variability of $NO_x$ in more detail, instruments on geostationary satellites are essential (Burrows et al., 2004). The South Korean instrument GEMS (Geostationary Environmental Monitoring Spectrometer, (Kim et al., 2020)), was launched in February 2020 and is the first instrument in geostationary orbit that delivers hourly daytime air quality observations, including $NO_2$. Positioned over the Equator at a longitude of 128.2°E, GEMS takes measurements with a spatial resolution of about $3.5\,km \times 8\,km$ over a large part of Asia. With up to 10 observations per day, GEMS can offer valuable insights into the

diurnal variability of $NO_2$ and other trace gases. NASA's TEMPO (Zoogman et al., 2017) launched in April 2023 and ESA's Sentinel-4 (Ingmann et al., 2012) planned for launch in 2024 will provide similar observations over North America and Europe, respectively.

A study by Kim et al. (2023) evaluated GEMS L2 v1.0 total $NO_2$ column data from November 2020 to January 2021 with four ground-based Pandora instruments, all located in Seosan, South Korea. They found correlation coefficients of 0.62-0.78 and

an underestimation of the ground-based $NO_2$ measurements by the GEMS data set. Even though these four sites are relatively close together, they show different diurnal variations of $NO_2$, indicating that local transport or emissions have a significant



influence. Zhang et al. (2023) evaluated their scientific POMINO-GEMS tropospheric $NO_2$ vertical column density (VCD) product with nine ground-based MAX-DOAS sites based on data from June-August 2021. The POMINO-GEMS product shows a modest correlation of 0.66 with the MAX-DOAS observations and a reasonable agreement of the observed diurnal variations but cannot achieve the much better correlation of 0.83 of the POMINO-TROPOMI product and the MAX-DOAS observations. Drivers of the diurnal variation of $NO_2$ observed by GEMS during winter and summer over Beijing and Seoul have been investigated by Yang et al. (2023b). They found good agreement between the diurnal variations of total $NO_2$ columns in Pandora, GEMS, and GEOS-Chem and used GEOS-Chem to interpret the observed variations. Due to high emissions at the two urban sites, $NO_2$ accumulates over the day, which is offset by losses from chemistry and transport depending on season and wind speed.

In this study, one year of tropospheric $NO_2$ VCDs of the operational GEMS L2 v2.0 product, the scientific GEMS IUP-UB v1.0 product, the operational TROPOMI product, and 11 ground-based DOAS instruments in South Korea are compared. Evaluating the GEMS $NO_2$ product is important to ensure the accuracy of the product for use in emission and surface concentration estimates (e.g., Xu et al., 2023; Yang et al., 2023c). The 11 ground-based observation sites are located in different pollution regimes in South Korea which provides the opportunity to observe and analyze different diurnal variations of $NO_2$. Including the TROPOMI product in the comparisons adds an already well-validated reference data set around noon. ECMWF reanalysis v5 (ERA5) wind data at 10 m altitude give valuable insights into the influence of transport effects on the diurnal variation. Using a full year of data allows to analyze the influence of seasonality and the weekday-weekend effect on the GEMS tropospheric $NO_2$ VCD.

The instruments and data sets included in this study are described in Sect. 2. After a first comparison of one month of averaged GEMS L2 v2.0, GEMS IUP-UB v1.0, and TROPOMI v02.04.00 tropospheric $NO_2$ VCDs maps in Sect. 3, one year of satellite observations is evaluated by comparisons with the tropospheric $NO_2$ VCD data set of the ground-based network distributed within South Korea (see Sect. 4) and car DOAS observations (see Sect. 4.2). In Sect. 5 the diurnal variations of the GEMS IUP-UB $NO_2$ product and the ground-based observations are analyzed. Influencing factors such as seasonality, wind speed, transport processes, and the weekday-weekend effect are evaluated to understand the observed diurnal variation. Possible reasons for deviations between GEMS and ground-based observations are discussed. A summary and conclusions are provided in Sect. 6.

## 2 Instruments and data sets

In this study, data from two measurement campaigns in South Korea are used: The GEMS Map of Air Pollution (GMAP) 2021 and the Satellite Integrated Joint Monitoring of Air Quality (SIJAQ) 2022 campaigns (https://www.sijaq.org). The main campaign periods were from October 2021 to November 2021 and May 2022 to August 2022. Some instruments were also operated between the main campaign periods and beyond. Analyses of this study focus on measurements taken between October 2021 and October 2022. One key aspect of these campaigns was to gather measurements for the validation and improvement of GEMS data, to better understand uncertainties and error sources in the satellite products, to support further improvement of



the satellite retrieval algorithms, and to apply GEMS data for the characterization of air pollution.

Instruments from several teams participated in the campaigns. Measurements were delivered by stationary MAX-DOAS and Pandora instruments, as well as mobile car DOAS instruments. Details on the different instruments are provided below in Sect. 2.3 and 2.4. During the GMAP 2021 campaign, measurements were focused on the Seoul Metropolitan Area (SMA), with a population of 26 million, one of the largest and most polluted metropolitan regions worldwide. During the SIJAQ 2022 campaign, measurements were additionally performed in the southern part of South Korea. This region includes Busan, the second largest city of South Korea, and Ulsan, an important industrial center. Figure 1 shows GEMS observations of tropospheric $NO_2$ vertical columns over South Korea, indicating several pollution hot spots and the locations of the stationary instruments. The combination of stationary and mobile measurements makes a comprehensive validation of GEMS data possible. The stationary MAX-DOAS measurements are located in different pollution regimes and provide daily measurements with good temporal resolution, but are restricted in spatial coverage. The car DOAS measurements lack temporal resolution but can cover larger areas of a satellite pixel and can be operated over several satellite pixels in different regions to cover a large variety of pollution levels. Table 1 lists all instruments involved in this study.

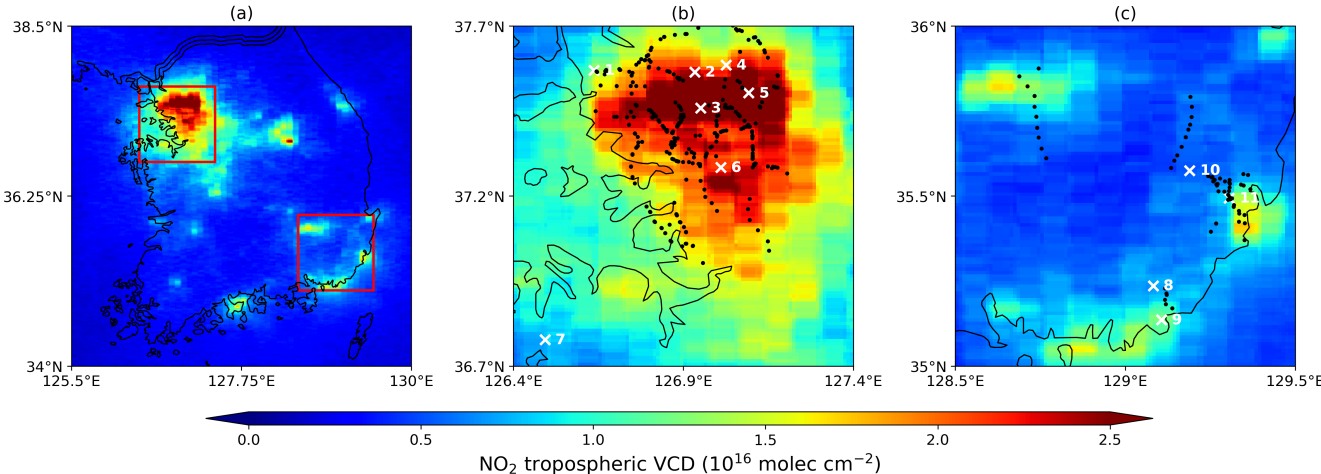

**Figure 1.** Maps of $NO_2$ tropospheric vertical columns for South Korea from GEMS IUP-UB v1.0 observations in October 2021 around 13:45 Korean Standard Time (KST) (04:45 UTC). Panel (b) is a zoom into the SMA region indicated by the upper red rectangle in panel (a). Panel (c) is a zoom into the southeast, indicated by the lower red rectangle in panel (a). The white crosses show the locations of the ground-based measurement sites. The different instruments are listed together with the number given in Table 1. Black dots indicate locations of car DOAS observations used for GEMS validation.

## 2.1 Geostationary Environmental Monitoring Spectrometer (GEMS)

GEMS is a step-and-stare UV-visible imaging spectrometer onboard the satellite GK2B (Geostationary Korea Multi-Purpose Satellite 2), launched into geostationary orbit on 18 February 2020. It is the first geostationary mission to monitor air quality



**Table 1.** List of instruments included in this study with location, observation geometry, VCD retrieval information, and period of observation. MAX-DOAS BIRA Seoul (4) and MAX-DOAS BIRA Suwon (6) sites are using the same instrument which was moved from Suwon to Seoul in December 2021.

| Instrument | Location/Platform | Observation geometry | VCD retrieval | Available data |
|---|---|---|---|---|
| GEMS | GEO-KOMPSAT-2B | | L2 v2.0 and IUP-UB v1.0 | 6-10 times/day |
| TROPOMI | Sentinel-5P | Push-broom, nadir | RPRO/OFFL v2.4.0 | 1-2 times/day (~13:30 KST) |
| MAX-DOAS IUP-UB Incheon (1) | Incheon (37.57° N, 126.64° E) | Multi-axis | FRM4DOAS 01.01 MMF | Oct 2021 - Oct 2022 |
| Pandora 54 Yonsei (2) | Seoul (37.56° N, 126.93° E) | Multi-axis | PGN rnvh3.1-8 | Oct 2021 - Oct 2022 |
| Pandora 149 SNU (3) | Seoul (37.46° N, 126.95° E) | Multi-axis | PGN rnvh1.1-7 | Oct 2021 - Oct 2022 |
| MAX-DOAS BIRA Seoul (4) | Seoul (37.59° N, 127.03° E) | Multi-axis | FRM4DOAS 01.01 MMF | Dec 2021 - May 2022 |
| MAX-DOAS MPIC Seoul (5) | Seoul (37.50° N, 127.09° E) | Multi-axis | FRM4DOAS 01.01 MMF | Oct 2021 - Aug 2022 |
| MAX-DOAS BIRA Suwon (6) | Suwon (37.28° N, 127.01° E) | Multi-axis | FRM4DOAS 01.01 MMF | Oct 2021 - Dec 2021 |
| Pandora 164 Seosan (7) | Seosan (36.78° N, 126.49° E) | Multi-axis | PGN rnvh3.1-8 | Oct 2021 - Oct 2022 |
| Pandora 20 Busan (8) | Busan (35.24° N, 129.08° E) | Multi-axis | PGN rnvh3.1-8 | Oct 2021 - Oct 2022 |
| MAX-DOAS MPIC Busan (9) | Busan (35.14° N, 129.11° E) | Multi-axis | FRM4DOAS 01.01 MMF | Jun 2022 - Aug 2022 |
| Pandora 150 Ulsan (10) | Ulsan (35.24° N, 129.19° E) | Multi-axis | PGN rnvh3.1-8 | Oct 2021 - Oct 2022 |
| MAX-DOAS IUP-UB Ulsan (11) | Ulsan (35.49° N, 129.31° E) | Multi-axis | FRM4DOAS 01.01 MMF | Jun 2022 - Oct 2022 |
| IUP car DOAS | Mobile car | Zenith-sky | | campaign based |
| MPIC car DOAS | Mobile car | Zenith and 22° | | campaign based |
| BIRA car DOAS | Mobile car | Zenith-sky | | campaign based |

hourly during the daytime. With its location at a longitude of 128.2°E over the Equator, GEMS covers a large part of Asia (5°S-45°S and 75°E-145°E). The ground pixels have a nominal resolution of approximately $3.5\,\mathrm{km} \times 8\,\mathrm{km}$ over Seoul. GEMS is operated in 4 scan modes and allows up to 10 observations per day over the eastern part of the covered area, including South Korea. Due to shorter days, the number of possible observations is reduced to 8 in March and October and is further limited to

a maximum of six observations in winter. The GEMS spectrometer covers the wavelength range of 300-500 nm with a spectral resolution of 0.6 nm. The measurements yield in column amounts of $O_3$, $NO_2$, $SO_2$, HCHO, CHOCHO, and also aerosol and cloud information (Kim et al., 2020). We use the tropospheric $NO_2$ VCD of the operational product and the scientific GEMS IUP-UB product, which are described below.

### 2.1.1 Operational GEMS L2 tropospheric $NO_2$ product v2.0

The operational GEMS L2 tropospheric $NO_2$ product v2.0 was reprocessed for the entire mission and is distributed by the National Institute of Environmental Research, NIER (https://nesc.nier.go.kr/en/html/cntnts/91/static/page.do). Data are available from 2021 onward. $NO_2$ slant column densities (SCDs) are retrieved based on a DOAS fit in a fitting window of 432-450 nm. Using a lookup table of altitude-dependent air mass factors (AMFs) and model based vertical profile shapes, the $NO_2$ SCDs are converted into $NO_2$ VCDs. In v2.0, the WRF-Chem + CAM-Chem model used in v1.0 (Lee et al., 2020), was replaced

with the GEOS-Chem model, which has a spatial resolution of $0.25\,° \times 0.3125\,°$. The altitude-dependent AMFs from the ra-





diative transfer model VLIDORT (Spurr, 2006) are tabulated as a function of the solar zenith angle (SZA), the viewing zenith angle (VZA), the relative azimuth angle (RAA), surface albedo, terrain height, temperature and pressure profiles, and aerosol parameters. The aerosol optical thickness (AOD), the single scattering albedo (SSA), and the aerosol layer height (ALH) are taken from GEMS L2 data. Since v2.0, the surface albedo is based on GEMS L2 surface reflectance data instead of the OMI

climatology. The cloud correction of the AMF uses a linear combination of a clear-sky and a cloudy AMF, weighted by the cloud radiance fraction. The separation of the total $NO_2$ VCD in its stratospheric and tropospheric parts is based on Bucsela et al. (2013), using GEOS-Chem model data for the tropospheric $NO_2$ column a priori and to mask high pollution regions.

To remove problematic retrievals and cloudy scenes, we use only observations with a final algorithm flag of 1 and a cloud fraction < 0.3. The product provides the 'root_mean_square_error' resulting from the $NO_2$ fit but does not include errors from

other retrieval aspects. Therefore, we are estimating the tropospheric $NO_2$ VCD error based on the assessment done for the TROPOMI product with a typical value over continental polluted areas of $\pm 25\%$, which is dominated by the uncertainties in the AMF calculation (van Geffen et al., 2022).

### 2.1.2 Scientific GEMS IUP-UB tropospheric $NO_2$ product v1.0

As part of the preparation for the European geostationary instrument on the satellite S4, a scientific GEMS $NO_2$ product has

been developed at the Institute of Environmental Physics at University Bremen (IUP-UB). The GEMS L1 spectra are analyzed with the DOAS technique in a larger fitting window from 405–485 nm and with corrections for instrument polarization sensitivity and scene inhomogeneity. The retrieved SCDs are corrected for the stratospheric contribution based on the STRatospheric Estimation Algorithm from Mainz (STREAM, (Beirle et al., 2016)). Tropospheric SCDs are converted into tropospheric VCDs with $NO_2$ a priori profile shapes from the TM5 chemical transport model (Williams et al., 2017) and a lookup table of altitude-

dependent AMFs computed with the radiative transfer model SCIATRAN (Rozanov et al., 2014). The TM5 model has a spatial resolution of $1° \times 1°$. The altitude-dependent AMFs are tabulated as a function of SZA, VZA, RAA, surface albedo, and surface height. The surface albedo is based on the TROPOMI Lambertian equivalent reflectivity (LER) climatology (Tilstra et al., 2023). To evaluate the influence of the surface albedo, an additional version was created using the GEMS L2 surface reflectance data. The AMF cloud correction is based on the independent pixel approximation and uses recalculated cloud fractions and

the cloud pressure from the GEMS L2 cloud product. The cloud fractions were computed from recalculated GEMS top of atmosphere (TOA) reflectances based on GEMS radiances and recalibrated irradiances by comparison with TOA reflectances modelled by SCIATRAN. In the current version of the algorithm, no aerosol correction is included. More details about the scientific GEMS IUP-UB tropospheric $NO_2$ v1.0 retrieval can be found in Richter et al. (in preparation, 2024).

Problematic retrievals and cloudy scenes with cloud radiance fractions of more than 50% are removed by using only obser-

vations with a qa_value above 0.75. The product contains the 'nitrogendioxide_tropospheric_vertical_column_density_uncertainty_random', which as in the operational product, only contains the random error from the fit. The tropospheric $NO_2$ VCD error is estimated based on the same $\pm 25\%$.



## 2.2 TROPOspheric Monitoring Instrument (TROPOMI)

TROPOMI is a hyperspectral imaging spectrometer onboard the sun-synchronous near polar-orbiting satellite Sentinel-5P (S5P), launched in October 2017 (Veefkind et al., 2012). With its measurements in the UV, visible, and IR spectral regions, TROPOMI can monitor several atmospheric trace gases as well as clouds and aerosols. We use the tropospheric $NO_2$ product retrieved from measurements in the visible channel (400-496 nm). The ground pixel sizes are approximately $3.5\,km \times 5.5\,km$ in the middle of the swath. With orbit times of around $100\,min$ and a wide swath of approximately $2600\,km$, TROPOMI has nearly global coverage and usually one to two overpasses per day in the mid-latitudes. Over the campaign region, TROPOMI provides observations between 12:28 and 14:40 Korean Standard Time (KST).

### 2.2.1 TROPOMI tropospheric $NO_2$ product v02.04.00

The latest TROPOMI tropospheric $NO_2$ product, reprocessed for the entire mission, is based on processor version 02.04.00. The v02.04.00 product was generated operationally from 17 June 2022 to 12 March 2023. We are using the offline (OFFL) as well as the reprocessed (RPRO) data of this version, which is available from the Sentinel-5P Pre-Operations Data Hub (last access: 21 February 2022). The following processor versions had only minor bug fixes and have not yet been applied to the full data set (Eskes and Eichmann, 2023). The Level 1b version 2.1 spectra are analyzed with the DOAS technique in a fitting window of 405-465 nm to retrieve $NO_2$ SCDs. The retrieved SCDs are separated into their stratospheric and tropospheric parts with $NO_2$ vertical profile information from the $1° \times 1°$ TM5 global chemistry transport model and a data assimilation system that assimilates TROPOMI SCDs. Using a lookup table of altitude-dependent AMFs and actual daily TM5 $NO_2$ vertical profile shapes, the resulting tropospheric SCDs are converted into tropospheric VCDs. The altitude-dependent AMFs are a function of SZA, VZA, RAA, surface albedo, surface pressure, and (mid-level) atmospheric pressure. Since v02.04.00, the surface albedo in the $NO_2$ spectral fitting window and in the cloud pressure retrieval is based on the TROPOMI directionally dependent LER (DLER) climatology (Tilstra et al., 2023). The cloud radiance fraction is retrieved from the $NO_2$ spectral region at 440 nm. The cloud pressure retrieval is based on the FRESCO-wide algorithm in the NIR spectral range. In the AMF, clouds and indirectly aerosol loads are accounted for using a linear combination of a clear-sky and a cloudy AMF, weighted by the cloud radiance fraction (van Geffen et al., 2022).

To remove problematic retrievals, we are using only observations with the recommended qa_value above 0.75. This also removes scenes with cloud radiance fractions in the $NO_2$ window of more than 50 % (Eskes and Eichmann, 2023). The TROPOMI $NO_2$ product contains the data field 'nitrogendioxide_tropospheric_column_precision', which provides the error estimate originating from the $NO_2$ fit and other retrieval aspects that is dominated by the uncertainty in the tropospheric air-mass factor (±25 %).

## 2.3 MAX-DOAS observations and data sets

The satellite tropospheric $NO_2$ VCDs are compared to collocated MAX-DOAS observations. We use data from MAX-DOAS instruments at six sites in South Korea, from which four were located in the northern campaign region and two in the southeast-



ern campaign region (see Table 1). Not all of them have been operated over the whole year. Data availability for satellite valida-
tion is also visible in Fig. 11. The ground-based MAX-DOAS instruments measure the UV-visible scattered sunlight in several
azimuthal directions and elevations. The here used tropospheric NO$_2$ VCDs are retrieved by applying the Mexican MAX-
DOAS Fit (MMF; (Friedrich et al., 2019)) inversion algorithm using the FRM4DOAS (v01.01, https://frm4doas.aeronomie.be/)
settings and setup (Hendrick et al., 2016). The product is quality filtered using only data with a recommended 'qa_flag_no2' of 0
and 1. This quality flag uses additionally the Mainz profile algorithm (MAPA, (Beirle et al., 2019b)) data in the quality check.
Details about the implemented algorithms and quality flagging approaches can be found in Hendrick et al. (in preparation,
2024). To ensure comparability between the MAX-DOAS instruments from different institutes, there was an inter comparison
period at the beginning of the campaign, during which all instruments except the MAX-DOAS IUP-UB Ulsan were operated
at the same location. The comparisons show very good correlations between the instruments. The inter-comparison results are
presented in Hendrick et al. (in preparation, 2024).

The Pandonia Global Network (PGN, https://www.pandonia-global-network.org) is a network of ground-based UV-visible
spectrometers called Pandora, which focuses on total column observations from direct sun measurements, but can also provide
tropospheric column observations when operated in multi-axis mode. We use data from the five Pandora instruments, located
in South Korea that are operated in the multi-axis mode. Three of them are located in the northern campaign region, and two
are in the southeastern region. Data are processed as part of the PGN Pandonia Global Network (last access: 4 February 2023).
All data except for the Pandora SNU are on the processor version 1.8 and retrieval version nvh3. Data of the Pandora SNU are
only available in processor version 1.7 with retrieval version nvh1. The most important update between the versions is a more
stringent quality filtering. We only use data with quality flags indicating high and medium quality and filter data of low quality
or which are flagged as unusable (flags 2, 12, 20, 21, and 22). Details on the retrieval of the tropospheric NO$_2$ VCD and the
respective uncertainty can be found in Cede (2021).

## 2.4 Car DOAS instruments and data sets

During the GMAP 2021 and SIJAQ 2022 main campaign periods, mobile car DOAS measurements were performed by in-
struments of the IUP-UB, the Max Planck Institute for Chemistry in Mainz (MPIC), and the Royal Belgian Institute for Space
Aeronomy (BIRA). To achieve high spatial resolution over the covered area, the majority of measurements was taken in zenith-
sky with some off-zenith measurements. Instruments were operated in both campaign regions and synchronized to the GEMS
schedule to cover several GEMS observations throughout the day. Compared to the stationary data, the car measurements have
the advantage that they can cover larger and more diverse areas. The car DOAS data analysis was done independently by the
operating institutes. More details about the car DOAS instruments and the tropospheric NO$_2$ VCD retrieval can be found in
Lange et al. (2023).





## 3    Satellite tropospheric NO$_2$ products comparison

Before comparing the satellite tropospheric NO$_2$ VCD products with ground-based measurements, an assessment of the three products based on maps of monthly averaged observations provides first insights into the differences between the two GEMS NO$_2$ products and the TROPOMI product. Figure 2 shows maps of NO$_2$ tropospheric VCD for South Korea from GEMS L2 v2.0 (a), GEMS IUP-UB v1.0 (b), and TROPOMI v02.04.00 (c) observations in October 2021. For better comparison, the two GEMS data sets are averaged only for the 13:45 KST (04:45 UTC) observation, which is close to the TROPOMI overpass between 12:28 KST and 14:37 KST. Data are sampled at 0.01° resolution.

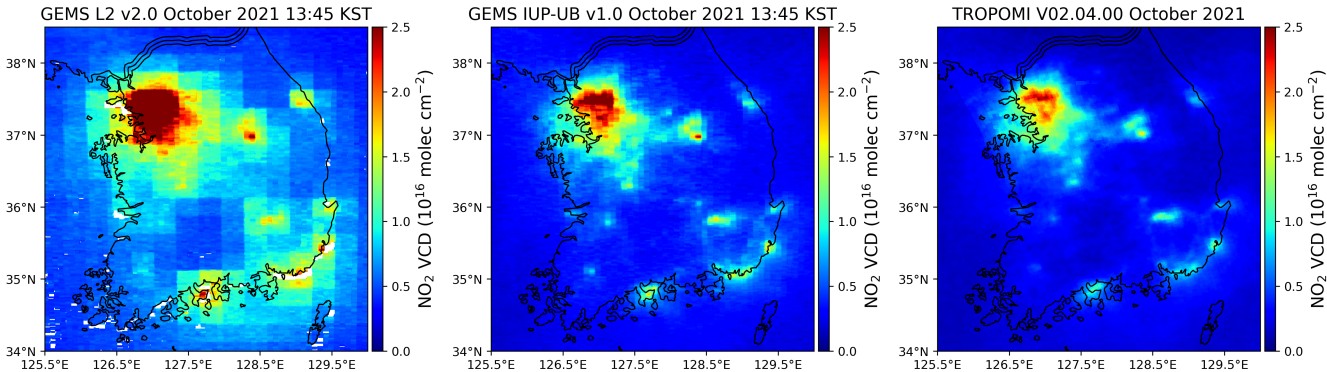

**Figure 2.** Maps of NO$_2$ tropospheric VCD for South Korea from GEMS L2 v2.0 (a), GEMS IUP-UB v1.0 (b), and TROPOMI v02.04.00 (c) observations in October 2021. The GEMS data sets are averaged for the 13:45 KST (04:45 UTC) observation close to the TROPOMI overpass. All data sets are cloud and quality filtered.

All three maps show the dominant hot spot of NO$_2$ centered over the SMA and several smaller hot spots with the Danyang county, including Jecheon and a mining area in the mid-north, Donghae on the east coast, Gwangyang in the south, and Daegu, Pohang, Ulsan, and Busan in the southeast. These hot spots show the highest values in the GEMS L2 tropospheric NO$_2$ VCD, especially over the SMA, followed by the GEMS IUP-Bremen product and the lowest values in the TROPOMI product. Additionally, we note that the background NO$_2$ is similar in the TROPOMI and GEMS IUP-UB products but significantly higher in the GEMS L2 product. This difference in the background NO$_2$ can be caused by the different stratospheric corrections used in the three products (GEOS-Chem with GEMS data assimilation, STREAM, and TM5 with TROPOMI data assimilation). The influence of the stratospheric correction will be visible most prominently in the evaluation with the ground-based data for stations located in remote regions such as the Pandora Ulsan (8, see Fig. 1). As the AMF is not interpolated in space, the map of the GEMS L2 v2.0 NO$_2$ product shows box structures with boxes of the same size as the spatial resolution of the GEOS-Chem model. The map of the TROPOMI NO$_2$ product appears the most smoothed, caused by the orbital cycle of 16 days and the resulting oversampling. Since GEMS maintains a constant ground pixel pattern for each of the four scan modes, there is no oversampling and smoothing, which makes the sampling pattern visible in the GEMS averages. Missing data in the GEMS L2 v2.0 NO$_2$ product, which are mainly visible in coastal regions, are caused by the product's quality filter.



# 4   Evaluating satellite tropospheric NO$_2$ VCD with ground-based data

The large data set of the ground-based instruments distributed in South Korea is used to evaluate the satellite tropospheric NO$_2$ VCD product. Ground-based data are averaged within ± 20 min of the satellite observation and compared to the closest pixels extracted within a radius of 5 km around the station sites. Scatter plots of all coincident measurements are shown in Fig. 3 for

the GEMS L2 (panel a), the GEMS IUP-UB (panel b), and the TROPOMI (panel c) NO$_2$ tropospheric VCD products. Since all 6-10 observations per day are considered for the comparisons of the GEMS products to the ground-based data set, there are 7928 coincident measurements for the GEMS L2 and 11823 for the GEMS IUP-UB product, which is many more than for the TROPOMI product with 1624. A comparison limited to the TROPOMI overpass time between 12:28 KST and 14:37 KST is shown in Fig. 4.  The difference in the number of coincident measurements between the GEMS L2 and the GEMS IUP-UB

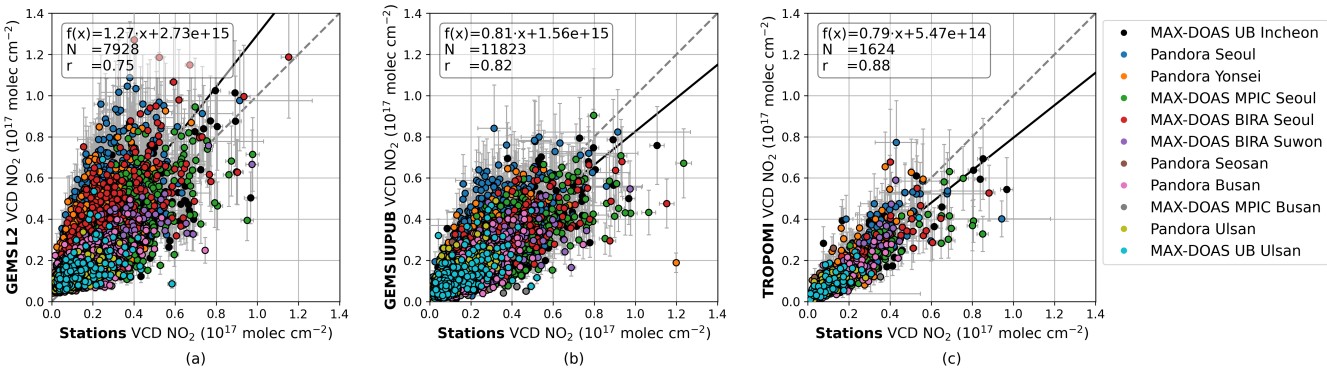

**Figure 3.** Scatter plots of GEMS L2 (a), GEMS IUP-UB (b), and TROPOMI (c) NO$_2$ tropospheric VCDs vs. co-located ground-based NO$_2$ tropospheric VCDs. The ground-based observations are considered co-located if they are taken within ± 20 min around the satellite observation. Measurements within this period are averaged and matched to the closest satellite observation within a radius of 5 km around the station site. The error bars represent the tropospheric NO$_2$ VCD error. Points are colored according to the corresponding ground-based instrument. The dashed gray line indicates the 1:1 line. The solid black line represents the orthogonal distance regression.

product is mainly caused by the stricter quality filter of the GEMS L2 product. Limiting the filter process of the GEMS L2 product on the cloud filter only, results in a more comparable number of data points.

All linear regression statistics in this study are calculated with orthogonal distance regression (ODR) to take into account the error in both evaluated and reference measurements. The correlation between the evaluated and reference measurements is described by the Pearson correlation coefficient r. Additionally, the median relative difference is calculated by the following

convention:

$$\text{median relative difference}(\%) = \frac{(\text{evaluated - reference})}{\text{reference}} \cdot 100 \tag{1}$$

The evaluated measurements are the satellite tropospheric NO$_2$ VCDs. The reference measurements are either the stationary ground-based or the mobile car DOAS tropospheric NO$_2$ VCDs. For satellite and ground-based matched, the coincidence criterion is described above. For the satellite and car DOAS coincidence criteria, see Sect. 4.2.



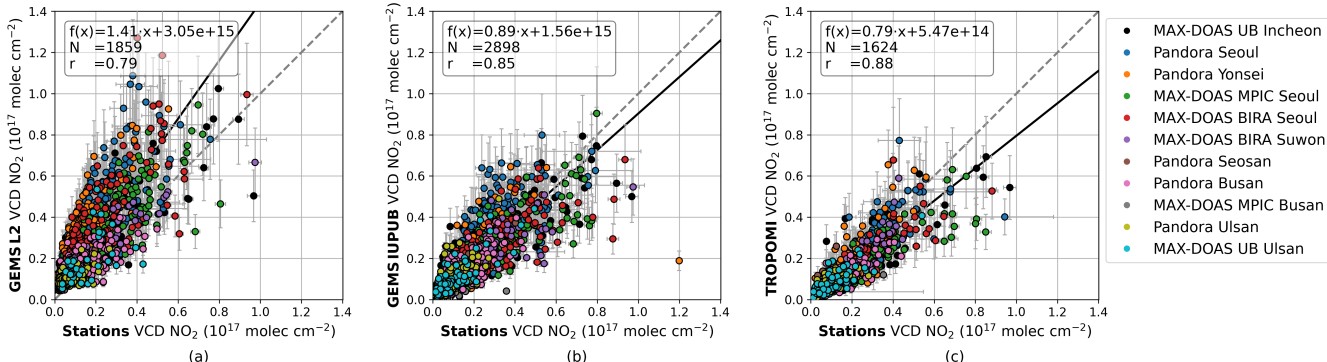

**Figure 4.** Same as Fig. 3 but GEMS L2 and GEMS IUP-UB observations are limited to the TROPOMI overpass time between 12:28 KST and 14:37 KST.

The GEMS L2 and ground-based tropospheric $NO_2$ VCDs are correlated with a Pearson correlation coefficient of r = 0.75, with a slope of 1.27, a median relative bias of +64 %, and an offset of $2.73 \times 10^{15}\,\mathrm{molec\,cm^{-2}}$. This overestimation is in contrast to the underestimation visible in the GEMS IUP-UB and TROPOMI $NO_2$ products. Potential explanations for this different bias, such as the surface reflectivity used for the AMF determination and the consideration of aerosol parameters, are further discussed in Sect. 5.4.

With a slope of 0.81 and a median bias of -1 % for the GEMS IUP-UB and a slope of 0.79 and a median bias of -14 % for the TROPOMI product, both products show a sight underestimation. This kind of underestimation of satellite $NO_2$ products compared to ground-based observations has been observed in many validation studies for satellite data sets (e.g., Ma et al., 2013; Verhoelst et al., 2021) and is often explained by local $NO_2$ hot spots that are not resolved in the satellite data and the a priori fields used for the AMF calculations. Another reason can be the missing aerosol correction in these satellite products.

When binning the median relative differences of the GEMS IUP-UB and MAX-DOAS comparison by the AOD, determined in the FRM4DOAS MAX-DOAS analysis, an increasing bias is observed with an increase in the AOD (see Appendix Fig. A1). This was similarly observed for a comparison of tropospheric $NO_2$ VCDs of MAX-DOAS and TROPOMI (Lambert et al., 2023).

    The GEMS IUP-UB product, considering all observations per day, has a good correlation with a coefficient of 0.82 but is more

scattered than the TROPOMI product, which is limited to its noon observation time. To investigate whether the better correlation of the TROPOMI product is attributed to the data itself or the timing of the satellite overpass, all data sets were limited to the period corresponding to TROPOMI overpasses. For the GEMS L2 product, this limitation amplifies the overestimation with a slope of 1.41 compared to 1.27 and a median bias of +64 % compared to +85 % but improves the correlation from 0.75 to 0.79. For the GEMS IUP-UB product, the slope and median bias increase from 0.81 to 0.89, respectively -1 % to +7 % and

brings the correlation of the GEMS IUP-UB product to 0.85, close to the very good correlation of the TROPOMI $NO_2$ product. This indicates larger deviations between the GEMS and MAX-DOAS observations in the morning and/or afternoon, which will be further analyzed by comparing the diurnal variability in Sect. 5.



The comparisons are based on coincident measurements considering the closest pixels within a radius of 5 km around the station sites. However, we have also compared the ground-based measurement with the closest pixels within a radius of 10 km,

averaging all pixels within a radius of 5 km, respectively 10 km, and considering the viewing azimuth angle (VAA) of the ground-based instruments to account for spatial inhomogeneity. The results are shown in Fig. A2 in the Appendix. To investigate the VAA dependence, the GEMS pixels $\mathrm{VCD_{sat}}$ are weighted according to their contribution along the line of sight $d$ of the ground-based instruments.

$$\mathrm{VCD_{sat,\,VAA}} = \frac{\sum \mathrm{VCD}_{\mathrm{sat}\,i} \cdot d_i}{\sum d_i} \qquad (2)$$

We consider the line of sight within 5 km to the station site. The comparison is only included in the analysis when more than 75 % of the line of sight is covered by satellite pixels. Since ground-based measurements taken within ±20 min of the satellite observation in different VAA are considered independently, this comparison results in more coincident data points. Measurements taken in the same VAA, overlapping with the same GEMS pixels, are averaged within the time window. Even if a better representation of spatial inhomogeneity is expected with this comparison, the results are either slightly worse or not

significantly better than the nearest pixel approach. This is the same for all three satellite products. Also, additional averaging of the VAA comparisons within the ±20 min time window does not improve the comparisons. This is in contrast to Dimitropoulou et al. (2020), who showed significant improvements in slope and correlation when considering the directional dependency for a comparison of TROPOMI and MAX-DOAS observations in Uccle, Belgium. Therefore, further investigations are required into why the comparisons in South Korea behave differently.

## 4.1 Comparison of satellite and ground-based tropospheric NO$_2$ VCDs for the individual sites

When separating the comparison of the satellite and ground-based observation into the individual sites, some differences can be observed between the different sites.

Figure 5 shows the scatter plots of GEMS IUP-UB tropospheric NO$_2$ VCDs vs. co-located ground-based NO$_2$ tropospheric VCDs for the 11 stations. The correlation varies between 0.67 for the MAX-DOAS IUP-UB Ulsan site and 0.87 for the

MAX-DOAS IUP-UB Incheon site. The slope varies between 0.40 for the MAX-DOAS MPIC Busan site and 1.17 for the Pandora SNU at Seoul National University (SNU). Scatter plots for the GEMS L2 and TROPOMI products can be found in the Appendix Fig. A3 and A4. For the GEMS L2, differences between the individual sites are even larger due to the dependence on the AMF box in which the station is located (see Fig. 2).

Figure 6 shows box-and-whisker plots for the three satellite NO$_2$ products and all stations summarizing the bias and spread

of the differences. The overall bias (median of all satellite and station pair differences) is +64 % for the GEMS L2 product, -1 % for the GEMS IUP-UB product, and -14 % for the TROPOMI product. A comparison by Lambert et al. (2023) based on tropospheric NO$_2$ VCDs of the Network for the Detection of Atmospheric Composition Change (NDACC) MAX-DOAS data from 29 stations and TROPOMI data of v2.4.0 and 2.5.0 from May 2018 to November 2023 shows a median bias of -28 % and for a subset of 8 MAX-DOAS stations in the TROPOMI Validation Data Analysis Facility Automated Validation Server

(VDAF-AVS) of -17.5 %. Thus somewhat larger than for the data set analyzed here.





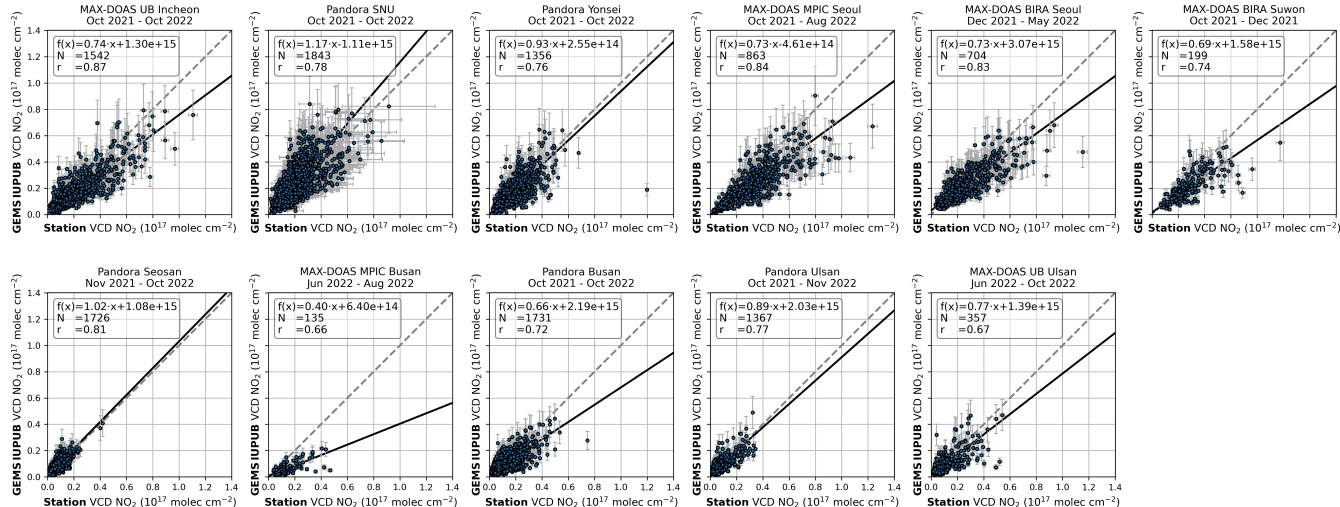

**Figure 5.** Scatter plots of GEMS IUP-UB tropospheric NO$_2$ VCDs vs. co-located ground-based NO$_2$ tropospheric VCDs for the 11 individual sites. Station names and measurement periods can be found in the title. Co-location criteria are with $\pm$ 20 min and nearest 5 km the same as in Fig. 3. Plots showing the comparisons for the GEMS L2 and TROPOMI products can be found in the Appendix Fig. A3 and Fig. A4.

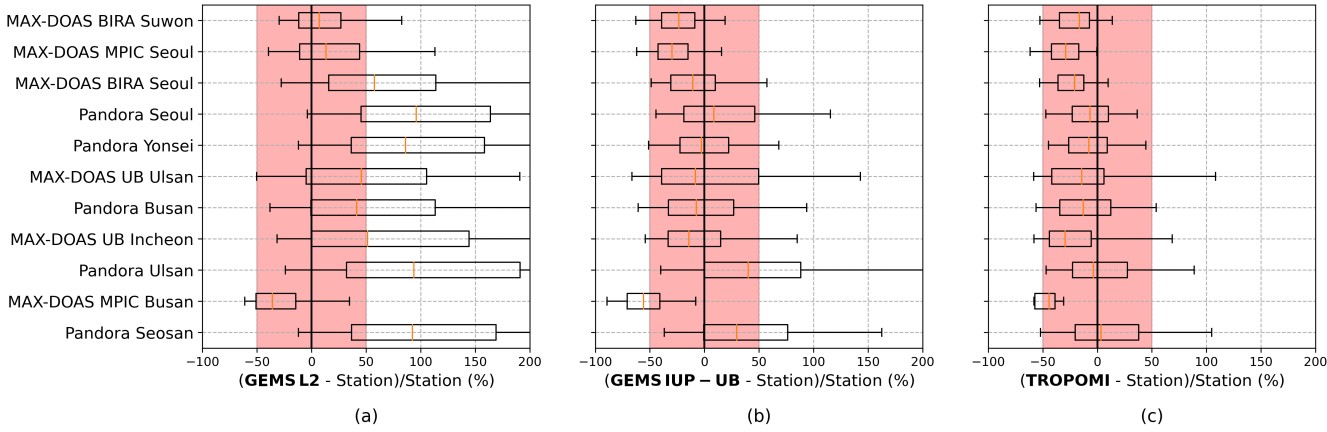

**Figure 6.** Box-and-whisker plots summarizing the bias and spread of the difference between the (a) GEMS L2, (b) GEMS IUP-UB, (c) the TROPOMI and the individual ground-based tropospheric NO$_2$ VCDs. Stations are ordered from bottom to top by increasing median ground-based tropospheric VCD. The orange line inside the box represents the median relative difference. Box bounds mark the 25 and 75 % quantiles. Whiskers represent the 5 and 95 % quantiles. The red shaded area represents a bias of $\pm$ 50 %. Plots with observations limited to the TROPOMI overpass time can be found in the Appendix Fig. A6.

For the GEMS IUP-UB and the TROPOMI product, the overall and the individual biases, except for the MAX-DOAS MPIC Busan, are within the typical mission requirement of a maximum bias of 50 % (van Geffen et al., 2022).

The large negative bias for the MAX-DOAS MPIC Busan site is visible in all product comparisons and is possibly caused by



its location close to the coast (<500 m) and the associated inhomogeneities. The sea-land breeze circulation can create complex
horizontal and vertical gradients in atmospheric composition, which are difficult to resolve in a priori profiles used for satellite
retrievals (e.g., Souri et al., 2023). Furthermore, the measurements of this instrument are performed in an azimuth direction of
$253°$, which crosses the port of Busan, a local source of $NO_x$. There is a slight tendency to larger biases for more polluted sites
while less polluted sites show differences closer to 0. These findings are similar to the validation results from Verhoelst et al.
(2021) on TROPOMI $NO_2$ data. The positive bias in the GEMS IUP-UB for the Pandora Ulsan and the Pandora Seosan, both
less polluted sites, could be an indication for an underestimation of the stratospheric contribution at these sites.

In general, it can be concluded that the GEMS IUP-UB product and the TROPOMI product show good agreement in the
individual biases, supporting the good agreement visible in the overall comparison. The agreement is improved when limiting
the GEMS IUP-UB product comparisons to the TROPOMI overpass time.

### 4.2   Comparison of GEMS IUP-UB and car DOAS observations

The car DOAS observations are used, in addition to the stationary ground-based observations, to evaluate the GEMS IUP-
UB tropospheric $NO_2$ VCDs. The IUP, MPIC, and BIRA car DOAS instruments were operated in the two campaign regions.
The locations of the car DOAS observations are displayed in Fig. 1. Compared to the stationary data, they can cover larger
and more diverse areas, reflected in the large range of $NO_2$ values shown in Fig. 7. The scatter plot shows GEMS IUP-UB

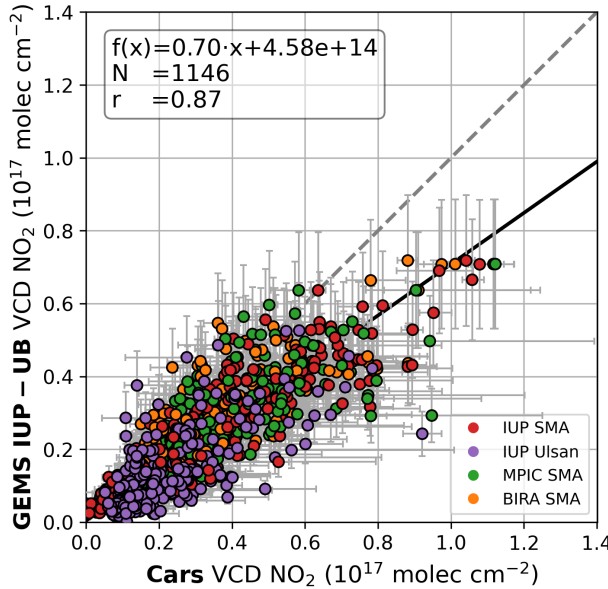

**Figure 7.** Scatter plot of GEMS IUP-UB tropospheric $NO_2$ VCDs vs. co-located car DOAS $NO_2$ tropospheric VCDs. The car DOAS
observations are considered co-located if they are taken $\pm$ 20 min around the GEMS observation within the satellite pixel. Each point is
colored by the respective car DOAS instrument. Vertical error bars represent the tropospheric $NO_2$ VCD error. Horizontal error bars are the
10th and 90th percentile to illustrate the spatiotemporal variability.



tropospheric $NO_2$ VCDs vs. co-located car DOAS $NO_2$ tropospheric VCDs. The car DOAS data are compared to the GEMS pixel in which they were measured, averaged $\pm$ 20 min around the GEMS observation. In total, 1146 pairs of coincident measurements are considered, of which 272 were taken during the TROPOMI overpass time window. The comparison between the GEMS IUP-UB and the car DOAS data shows a good correlation with a correlation coefficient of 0.87. Thus, they are better correlated than the stationary ground-based data with a correlation of 0.82. The slope of 0.7 and a median relative bias of -30 %, indicates a larger negative bias than the comparison with the stationary ground-based data set. This larger underestimation of the GEMS IUP-UB product may be caused by the bias of the larger proportion of high $NO_2$ observations, which was already visible from the evaluation by individual stations for the more polluted sites. Considering that the car DOAS data used for this comparison were analyzed independently by the different groups and with only partly harmonized retrieval methods with different assumptions, the data show good agreement, providing an additional data set for the evaluation of GEMS data. Horizontal error bars represent the 10th and 90th percentile of car DOAS observations within the GEMS pixel and $\pm$ 20 min time intervals to illustrate the spatiotemporal variability. These error bars can become relatively large, indicating the considerable temporal and spatial natural variability of $NO_2$ even within the GEMS pixel. Further investigations based on the car DOAS observations can provide more insights into the representativeness of observations and the natural variability.

## 5 Diurnal variation of GEMS and ground-based tropospheric $NO_2$ VCDs

As GEMS is the first geostationary instrument able to observe the diurnal variation of $NO_2$, it is interesting to compare the diurnal variation found in the GEMS data with those observed by the ground-based instruments. In Fig. 8, the median diurnal variation of the GEMS IUP-UB product and the ground-based station data are shown for 10 of the 11 stations. The diurnal variability of tropospheric $NO_2$ VCDs from the GEMS L2 v2.0 product and ground-based observations can be found in the Appendix Fig. A5. Due to limited data availability at the MAX-DOAS MPIC Busan site, it is not shown here. Also, the MAX-DOAS BIRA Suwon and the MAX-DOAS IUP-UB Ulsan sites have limited data due to their relatively short operation time, indicated in the green bar at the top of each plot. However, eight sites have a large enough data set, which can be compared to the diurnal variability seen in the GEMS data. The overall behavior of the GEMS IUP-UB and the ground-based data are very similar. However, the two most polluted sites, the MAX-DOAS MPIC Seoul and the MAX-DOAS BIRA Suwon show a clear low bias of the GEMS IUP-UB data, already seen in Fig. 6. This underestimation is visible throughout the whole day but is largest in the morning. The low-polluted sites, the Pandora Seosan and the Pandora Ulsan show a high bias of the GEMS IUP-UB data, which is relatively stable over the day. The other sites show good agreement, especially around noon from around 11 to 14 KST. Deviations are visible in the morning and afternoon, where the GEMS IUP-UB product often underestimates the station values, which is most prominently for the MAX-DOAS UB Incheon, the Pandora Yonsei, and especially the MAX-DOAS BIRA Seoul site. These biases are summarized in Fig. 9, which shows the diurnal variability of the median relative differences of the GEMS IUP-UB $NO_2$ product at the individual sites. This shows, as discussed, the best agreement around noon and the larger biases in the morning and late afternoon, especially for the 16:45 KST observation where the GEMS IUP-UB $NO_2$ product underestimates all sites observations except for the Pandora Seosan.



**Figure 8.** Diurnal variability of the median tropospheric NO$_2$ VCDs from the GEMS IUP-UB product (blue) and ground-based stations (red). The TROPOMI observation is added in black. Error bars represent the 25 and 75 % quantiles of the MAX-DOAS and GEMS observations. The numbers in the green bar represent the number of GEMS and MAX-DOAS observations that contributed to the median value. Station names and measurement periods can be found in the individual titles.





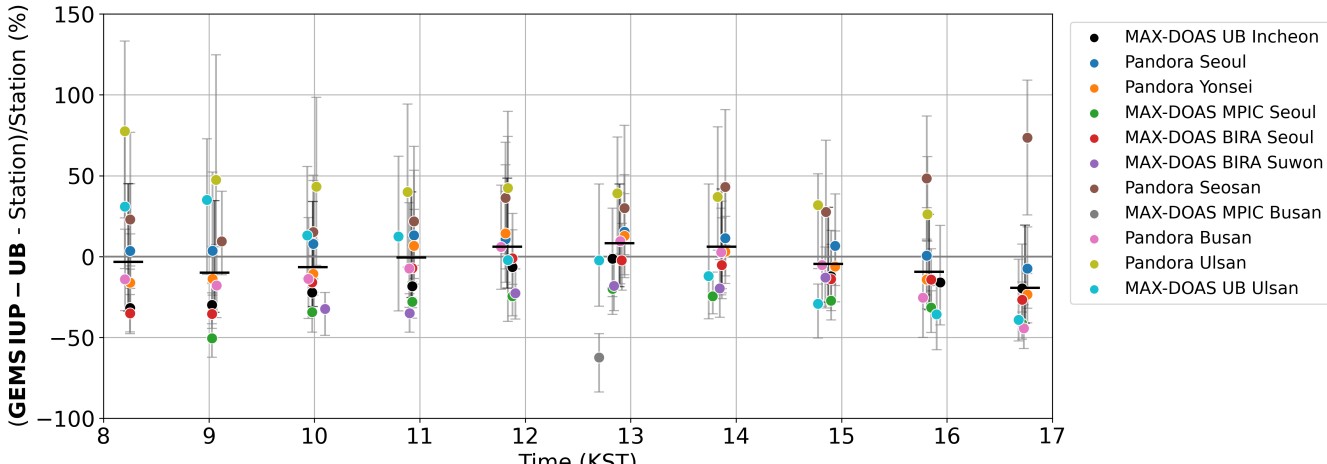

**Figure 9.** Diurnal variability of the median relative differences of the GEMS IUP-UB NO$_2$ product at the different ground-based sites. Stations are color-coded. The median relative difference, including differences of all stations, is shown as black bars. Error bars represent the 25 and 75 % quantiles. Results are only included if more than 20 observations are available per time bin and station.

It is interesting to see that the diurnal variation can be quite variable between the different sites. The Seoul stations (Pandora SNU, MAX-DOAS MPIC, MAX-DOAS BIRA) show quite similar diurnal variations with increasing NO$_2$ in the morning, a maximum around 12 KST, and a decrease towards the evening. In general, this aligns with previous studies, which found up to 40 % reduction of NO$_2$ columns in the OMI afternoon overpass compared to the SCIAMACHY morning over urban regions

(Boersma et al., 2008, 2009) and similar reductions using GOME-2 morning and OMI afternoon observations over large urban regions in the US (Penn and Holloway, 2020). However, the GEMS observations reveal that the morning observations of SCIAMACHY and GOME-2 are in the increasing part, while the afternoon observations of OMI are in the decreasing part, and the maximum of NO$_2$ in Seoul around 12 KST is in between and not captured by previous missions.

The data sets at the Pandora Yonsei site and the MAX-DOAS UB Incheon, which already is at the edge of the SMA, show less

diurnal variation with an earlier maximum around 11 KST, a slight decrease, and more of a plateau in the afternoon. The less polluted sites show little diurnal variability. For the Pandora Seosan site, the NO$_2$ slightly decreases over the day towards the evening. For the Pandora Ulsan site, on the other hand, NO$_2$ increases slightly over the day.

Interpreting these observed differences in diurnal variability is difficult as they are driven by emissions, chemistry of NO$_x$, and transport processes. These driving factors vary with season, wind speed, transport processes, and weekday-weekend effects,

which are analyzed in more detail in the following sections.

## 5.1 Seasonality

Figure 10 shows the diurnal variability divided into winter (DJF), spring (MAM), summer (JJA), and autumn (SON) from the GEMS IUP-UB and the station's data sets for those stations which were operated over the whole year. In general, NO$_2$ values





**Figure 10.** Diurnal variability of median tropospheric NO$_2$ VCDs from the GEMS IUP-UB product (blue) and ground-based stations (red) for the individual seasons (DJF, MAM, JJA, SON). The TROPOMI observation is added in black. Station names can be found in the individual titles. Error bars represent the 25 and 75 % quantiles of the MAX-DOAS and GEMS observations.

are higher in winter than in summer. For summer months (JJA), several stations (MAX-DOAS UB Incheon, Pandora Yonsei,

Pandora SNU) show a minimum in the NO$_2$ VCDs around noon (around 13 KST). This observation would fit the expectation that chemical loss is strong during noon, especially in summer, and significantly influences the diurnal variability of NO$_2$. However, this summer noon minimum is more pronounced in the stationary observations than in the GEMS IUP-UB columns, which show less diurnal variation. Based on the less significant role of NO$_x$ chemistry during winter time and expected higher emissions, one would expect an accumulation of NO$_2$ and increasing NO$_2$ VCDs over the day. This was also seen by Yang et al.

(2023b) for total column observations of GEMS and two Pandoras in Beijing and Seoul. However, this is not visible for the five sites analyzed here. On the contrary, the observed tropospheric NO$_2$ VCDs tend to decrease over the day. Most variation



over the day is visible for spring and autumn months. During these seasons, the NO$_2$ follows the described variability with an increase during the morning, a maximum at late morning, respectively noon, and a decrease towards the afternoon. Since these months have the highest data availability, they are dominating the diurnal variability averaged over the whole observation

period, seen in Fig. 8.

Differences between the GEMS IUP-UB and the station columns differ between seasons and sites. For the Pandora SNU site, the GEMS IUP-UB product overestimates the station columns in autumn, but both show very similar diurnal variability. In summer, on the other hand, the GEMS IUP-UB product underestimates the station columns, especially in the morning and afternoon. A similar behavior is visible for the MAX-DOAS UB Incheon and Pandora Yonsei sites. The Pandora Seosan site

shows the overestimation of the GEMS IUP-UB product, already seen in the overall diurnal variation plot, in all seasons. The differences seen for the Pandora Busan site in the late afternoon are dominated by the spring and summer observations.

These differences are also summarized in a heatmap plot in Fig. 11, showing an underestimation of the GEMS IUP-UB compared to the station tropospheric NO$_2$ VCDs in blue and overestimation in red. In general, no overall seasonality of the

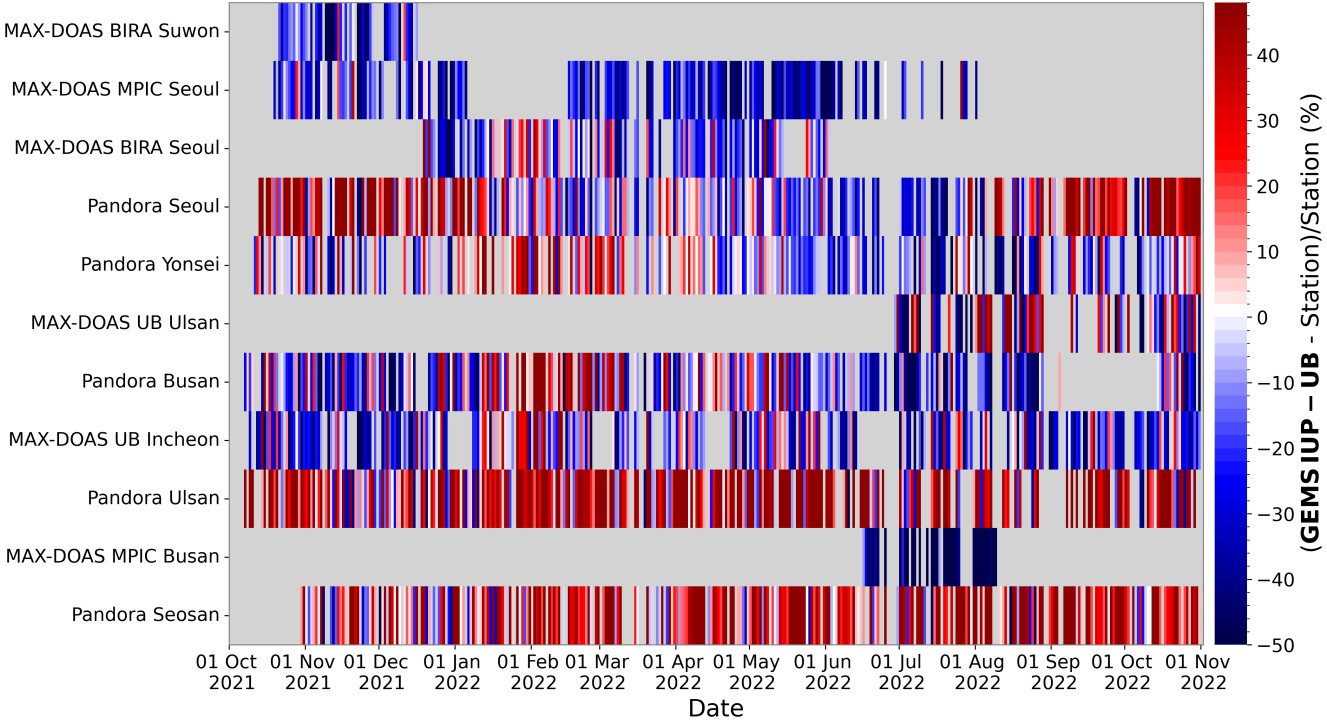

**Figure 11.** Time series of the median relative differences at the different ground-based sites from 1 October 2021 to 31 October 2022. The stations are ordered from bottom to top by increasing median ground-based tropospheric VCD.

biases is visible. For the Pandora SNU site, the discussed positive bias is visible for the autumn and winter observations. For

the Pandora Busan site, the GEMS IUP-UB product overestimates from the beginning of January to the beginning of March





while having mostly negative biases for the rest of the year. Additionally, the already discussed overall positive bias for the Pandora Ulsan and Seosan sites is visible, which seems to be a bit more pronounced in spring.

## 5.2  Effects of wind speed and transport processes

Figure 12 and Fig. 13 illustrate the sensitivity of the diurnal variability of $NO_2$ to wind speed. Observations are separated into calm (wind speeds $< 3\,\mathrm{ms^{-1}}$, Fig. 12) and windy (wind speeds $\geq 3\,\mathrm{ms^{-1}}$, Fig. 13) conditions based on ERA5 10 m wind data (Hersbach et al., 2023), temporally and spatially interpolated to the GEMS observations. Due to reduced data availability after the separation, only selected sites are shown.  The diurnal variability is quite different for calm and windy conditions for some

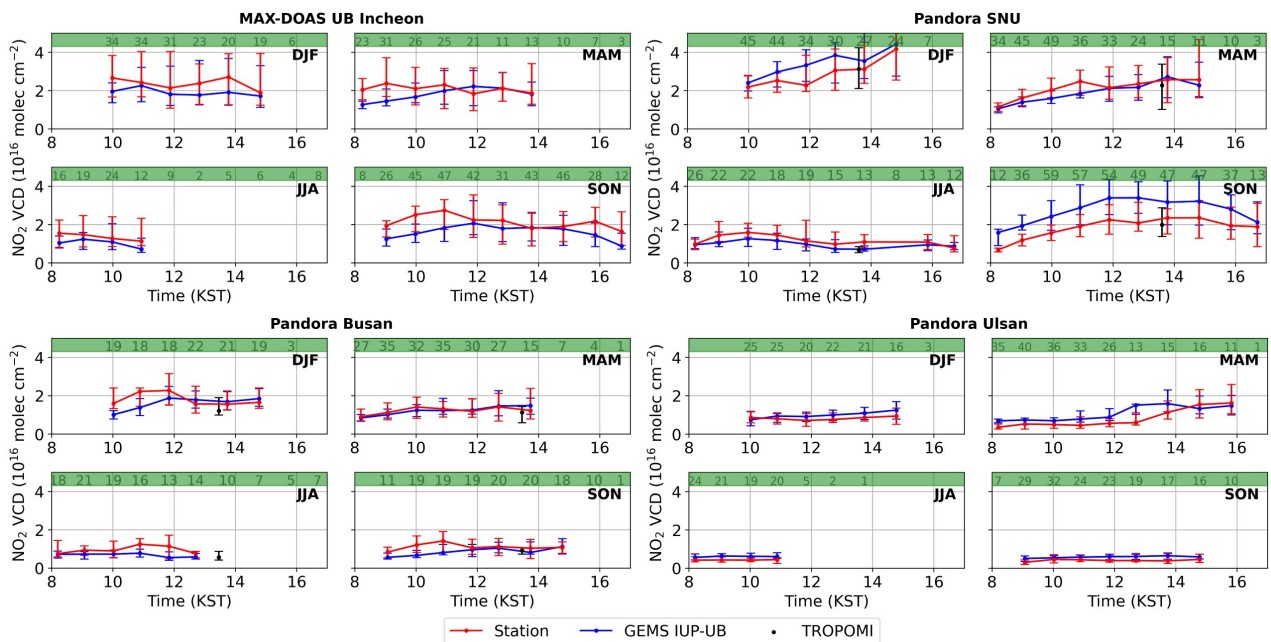

**Figure 12.** Same as Fig. 10 but only including observations with wind speeds $< 3\,\mathrm{ms^{-1}}$ for selected sites with still good data availability.

of the shown sites and seasons but consistent for the GEMS IUP-UB and the ground-based data. However, the agreement is better for windy conditions than for low wind speeds, which can be explained by more dispersion during windy conditions, resulting in less inhomogeneities. For calm conditions, $NO_2$ columns are generally larger due to accumulation of emissions. For windy conditions, the observations show much less variability over the day because emissions are dispersed quickly. Largest differences between calm and windy conditions and between the seasons are found for the Pandora SNU, the most polluted site of the four. During calm days in winter, the $NO_2$ shows a strong increase over the day. This can be attributed to the less effective chemical loss in winter and the accumulation of emissions that cannot be balanced by dilution on calm days. This increase over the day in winter was also shown based on GEMS total $NO_2$ columns by Yang et al. (2023b) but already when considering all data, which is not visible in our tropospheric $NO_2$ VCD data set. After filtering for windy conditions





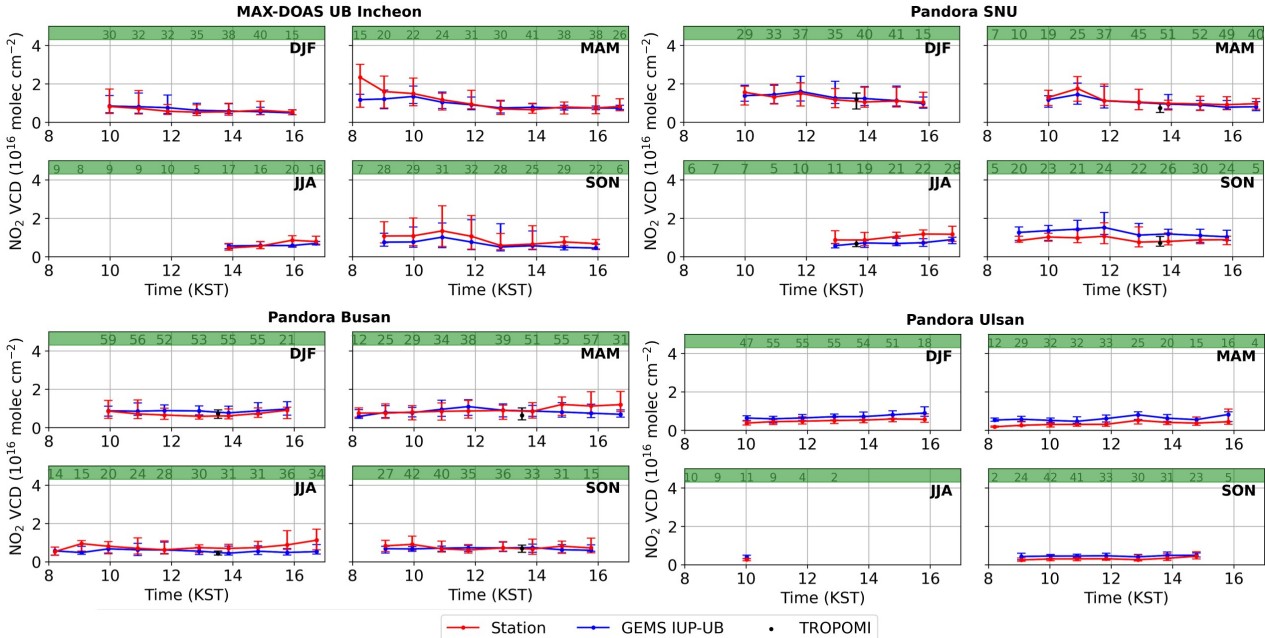

**Figure 13.** Same as Fig. 10 but only including observations with wind speeds $\geq 3\,\mathrm{ms}^{-1}$ for selected sites with still good data availability.

they observed an even stronger increase. For spring and autumn, the NO$_2$ increases in the morning, decreases around noon and flattens out in the afternoon. During summer, when chemical loss is even more effective, the minimum is around noon, but in general there is less variability.

Differences between calm and windy conditions are smaller for the less polluted sites. The Pandora SNU is the only site showing a strong increase over the day in winter during calm conditions.

Interestingly, a significant increase is visible in the Pandora Ulsan site observations in spring during low wind speeds. The increase is not happening over the whole day but starting around noon. This can be explained by transport effects, as illustrated by Fig. 14.

Figure 14 shows maps of GEMS IUP-UB tropospheric NO$_2$ VCDs averaged for May 2022 for each of the ten observations per day. Overlaid are the interpolated ERA5 10 m wind data. The maps show the southeast of South Korea, including the sites of the Pandora Busan, the MAX-DOAS MPIC Busan, the Pandora Ulsan, and the MAX-DOAS Ulsan. The GEMS IUP-UB NO$_2$ columns are highest for the late morning observations and are decreasing towards the evening. Interesting is the varying location of the NO$_2$ maximum over the day, clearly visible along the coastline and from its location relative to the station sites.

In the early morning, the NO$_2$ is mainly located at the MAX-DOAS UB Ulsan site. With the wind turning from mainly westerly in the morning to mainly southerly winds around noon, the NO$_2$ is moving northwards. Therefore, the NO$_2$ is moving closer to the Pandora Ulsan site, which can probably explain the increase starting around noon visible in the spring diurnal variation plot.





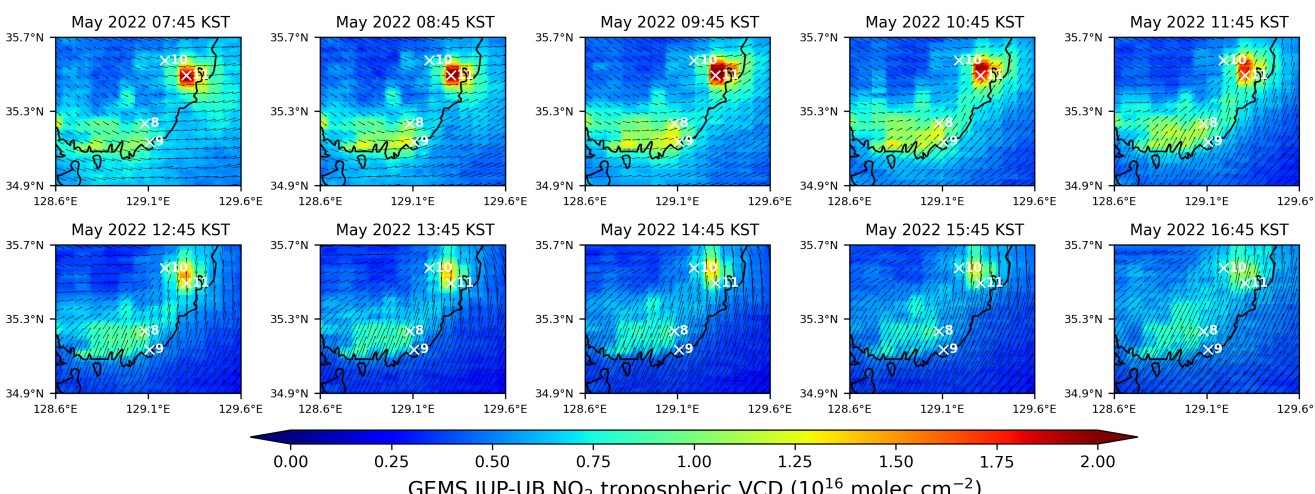

**Figure 14.** Maps of GEMS IUP-UB tropospheric NO$_2$ VCDs for the ten observations per day averaged for May 2022 overlaid with ERA5 10 m wind data. Maps show the southeast of South Korea, including the sites of the Pandora Busan (8), the MAX-DOAS MPIC Busan (9), the Pandora Ulsan (10), and the MAX-DOAS Ulsan (11).

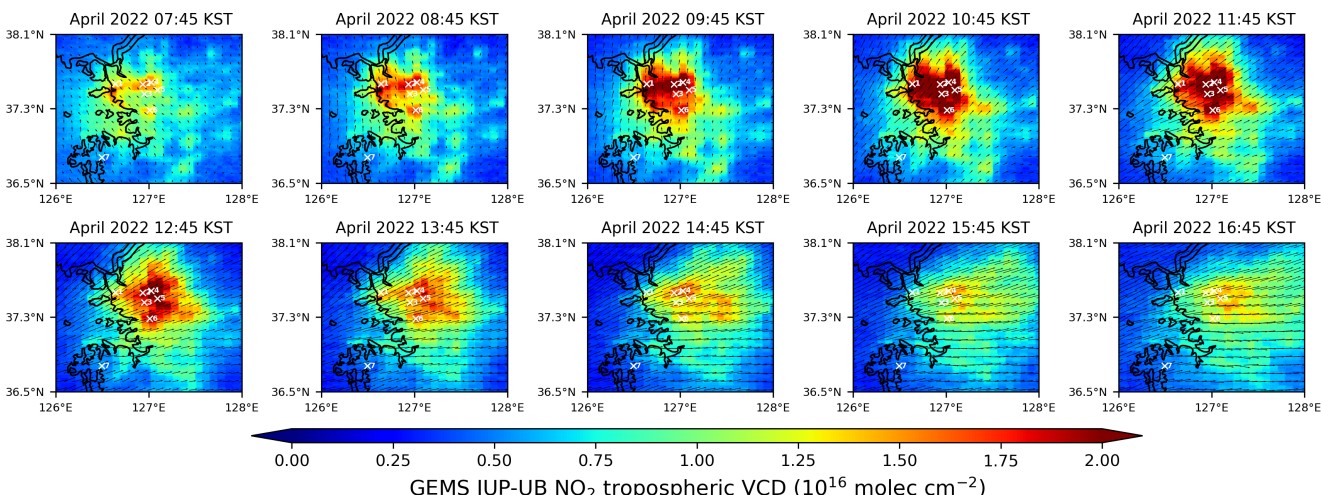

**Figure 15.** Maps of GEMS IUP-UB tropospheric NO$_2$ VCDs for the ten observations per day averaged for April 2022 overlaid with ERA5 10 m wind data. Maps show the SMA, including the sites of the MAX-DOAS IUP-UB Incheon (1), Pandora Yonsei (2), Pandora SNU (3), MAX-DOAS BIRA Seoul (4), MAX-DOAS MPIC Seoul (5), MAX-DOAS BIRA Suwon (6), and Pandora Seosan (7).

Figure 15 illustrates a similar variation for the SMA. The maps show the GEMS IUP-UB tropospheric NO$_2$ VCDs averaged for the ten observation times per day for April 2022 over the SMA with overlaid ERA5 wind data. The GEMS IUP-UB NO$_2$ columns are increasing during the morning, reaching their maximum for the 10:45 KST observation, and are decreasing towards





the evening. While in the early morning, the NO$_2$ is concentrated over the SMA's central part with most of the instruments, it is moving eastwards, with the wind direction turning from easterly low wind speeds to strong westerly winds. For the 12:45 and 13:45 KST observations, the NO$_2$ hot spot is located east of the Pandora Yonsei and SNU sites.

Additionally to the diurnal variability of transport effects due to changing wind direction, Fig. 16 illustrates also the seasonal variability. Shown are maps of monthly averaged GEMS IUP-UB tropospheric NO$_2$ VCDs for the 13:45 KST observation from October 2021 to September 2022 for the southeast of South Korea, with overlaid ERA5 10 m wind data. The GEMS

**Figure 16.** Maps of monthly averaged GEMS IUP-UB tropospheric NO$_2$ VCDs for the 13:45 KST observation from October 2021 to September 2022 overlaid with ERA5 10 m wind data. Maps show the southeast of South Korea, including the sites of the Pandora Busan (8), the MAX-DOAS MPIC Busan (9), the Pandora Ulsan (10), and the MAX-DOAS Ulsan (11).

IUP-UB NO$_2$ columns are highest from late autumn to early spring and have their minimum during the summer months. From September to January, with a mainly northwesterly wind direction, a large part of the NO$_2$ is located over the ocean and mostly
south of the ground-based stations. During spring, when the wind is changing from northwest to mostly southwest, the NO$_2$ is moving northwards.





The described influences of wind speed, causing dispersion or accumulation, and transport effects due to varying wind directions over the day and the year complicate the interpretation of observed diurnal variations of tropospheric $NO_2$ VCDs in terms of emissions and chemistry.

### 5.3 Weekday-weekend effect

Another influence on the diurnal variation of $NO_2$ is the difference in emissions for working days and weekends. Figure 17 shows the $NO_2$ VCDs of the day of the week normalized with the mean $NO_2$ from Monday to Friday for the GEMS IUP-UB, the GEMS L2, and the TROPOMI observations with the collocated station observations. GEMS observations are averaged over all available observations per day. TROPOMI observations are only available once or twice a day in cloud-free conditions. Therefore, some deviations between the TROPOMI and GEMS observations can be explained due to the reduced data availability or the timing effect. Due to different sampling for the three satellite products, each has its own coincident stationary data set. Generally, there is a good agreement between the respective satellite products and their corresponding ground-based measurements. Similarly, there is a good agreement among the different products, with few exceptions, mainly caused by the sparse TROPOMI observations. During the weekdays, from Monday to Friday, most normalized VCDs are close to one. Already on Saturday, the $NO_2$ is reduced compared to the weekdays. On Sundays, the $NO_2$ is reduced between around 20 % and 50 % compared to the average observed on weekdays. This reduction is significantly larger than the 10-20 % found in studies based on OMI and GOME data over Seoul (Beirle et al., 2003; Stavrakou et al., 2020). The smallest decline over the weekend is observed in Seosan, a more remote station with less influence of traffic emissions. For some sites, $NO_2$ is already reduced on Fridays, i.e., Pandora Busan. MAX-DOAS BIRA Seoul and MAX-DOAS MPIC Seoul are, for example, both located in Seoul still show some differences. $NO_2$ at the MAX-DOAS BIRA Seoul site peaks on Fridays and shows similarly strong reductions on Saturdays and Sundays, while at MAX-DOAS MPIC Seoul $NO_2$ peaks on Thursdays, and the reductions are strongest on Sundays. This could be due to local differences but also to the different months in which the stations are operated and the data analyzed. Large differences in the TROPOMI observations compared to the GEMS observations, e.g., Fridays for the MAX-DOAS BIRA Seoul site, could be explained by the different sampling with observations between 12:28 KST, and 14:37 KST, which might be biased to certain weeks or months because of cloud cover.

Interesting to see is the deviation of the GEMS L2 product on weekends. The agreement with the other data sets is very good during the weekdays, but on Saturday and Sunday, there is less reduction compared to the average observed on weekdays than the other products. One possible explanation is the in general higher background $NO_2$ values in the GEMS L2 product, which do not have a weekly cycle.

Plots of the diurnal variability on the weekend and weekdays from the tropospheric $NO_2$ VCDs of the GEMS IUP-UB product and the ground-based stations can be found in the Appendix Fig. A7.







**Figure 17.** Plots of normalized weekday NO$_2$ VCDs for the co-located station observations with the GEMS IUP-UB (blue), the GEMS L2 (cyan), and the TROPOMI (turquoise) observations. The corresponding station measurements are marked in pink for the GEMS L2 product, in red for the GEMS IUP-UB product, and in yellow for the TROPOMI product. Station names and operation periods can be found in the individual titles.





## 5.4 Discussion of GEMS - ground-based deviations

The agreement between the GEMS IUP-UB and the ground-based observations is already very promising. However, possible
explanations for observed differences have to be discussed.

One potential reason for deviations between the GEMS and ground-based observations could be a poor stratospheric correction. Since the contribution of the stratosphere is small with column densities in the order of $10^{14}\,\mathrm{molec\,cm^{-2}}$, especially compared to the typically observed tropospheric NO$_2$ VCDs in the range of $10^{16}\,\mathrm{molec\,cm^{-2}}$, it is not very likely. However, under some conditions, the operational GEMS product has large bias in the stratospheric columns, and in such situations, the stratospheric
correction can be a significant source of error.

Another explanation could be the effect of the Bidirectional Reflectance Distribution Function (BRDF), especially for the observations of diurnal evolution. However, the GEMS L2 product considers the BRDF influence using GEMS reflectivity data, yet the discrepancy with the ground-based data remains. To investigate the BRDF effect on the GEMS IUP-UB product, we replaced the TROPOMI LER product, used in the GEMS IUP-UB product, with the GEMS L2 reflectivity. Figure 18
shows the scatter plots of (a) the GEMS L2 product using the GEMS L2 reflectivity in the AMF calculation, (b) the GEMS IUP-UB product using the TROPOMI LER reflectivity, both analyzed before, and (c) the modified GEMS IUP-UB product using the GEMS L2 reflectivity. The modified GEMS IUP-UB product shows more scatter than the original version and an

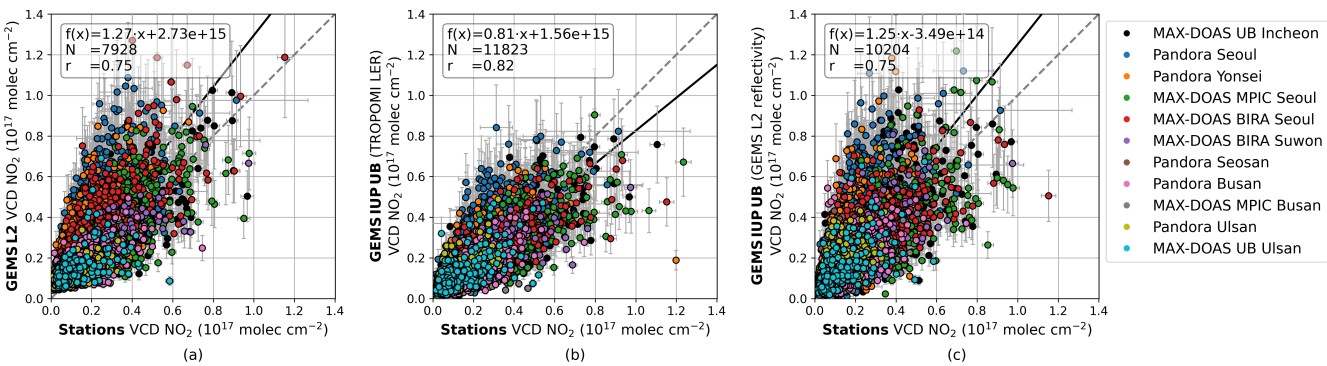

**Figure 18.** Scatter plots of satellite vs. co-located ground-based NO$_2$ tropospheric VCDs. For (a) the GEMS L2 product using the GEMS L2 reflectivity in the AMF calculation, (b) the GEMS IUP-UB product using the TROPOMI LER reflectivity, and (c) the GEMS IUP-UB product using the GEMS L2 reflectivity.

overestimation similar to the GEMS L2 product indicating that the GEMS reflectivity causes a large part of the overestimation and scatter.
Discrepancies between the MAX-DOAS and GEMS IUP-UB observations could be explained by the used model profiles. With its resolution of $1° \times 1°$, the TM5 model has a rather poor spatial resolution compared to the GEMS pixel size and the spatial variability of NO$_2$. Furthermore, it should be noted that the TM5 model has no specific focus on the GEMS region. Yang et al. (2023a) demonstrates that an updated version of the GEOS-Chem standard model with a resolution of $0.25° \times 0.3125°$




well reproduces diurnal variation of NO$_2$ vertical mixing observed during the KORUS-AQ campaign. This could be further
investigated by considering MAX-DOAS profiles and the GEMS averaging kernels. However, car DOAS measurements show
that sometimes there can be large fluctuations within individual satellite pixels, and station measurements may be located in
sub-pixel regions that are not representative of the entire pixel.

Another already mentioned aspect possibly contributing to the differences, especially at larger SZA, is the lack of knowledge
of tropospheric aerosol in the calculation of the AMF for the GEMS IUP-UB product. However, the L2 product considers
aerosol parameters from GEMS observations in the AMF determination and should correct for their influence. The expected
improvement is not reflected in the comparisons.

Due to less sensitivity at higher SZA (and VAA), AMFs are expected to be more uncertain for these scenes. This uncertainty
is further enhanced for larger aerosol loads and with low boundary layer heights in the morning and evening.

## 6 Summary and conclusions

We evaluated tropospheric NO$_2$ VCDs of the operational GEMS L2 v2.0, the scientific GEMS IUP-UB v1.0, and the oper-
ational TROPOMI v02.04.00 product with ground-based DOAS observations from 11 stationary and additional mobile car
DOAS instruments in South Korea. GEMS is the first instrument in geostationary orbit, enabling the observation of diurnal
variations of NO$_2$ for a large part of Asia. With its location centered over South Korea, GEMS provides up to 10 observations
during daytime. The GEMS IUP-UB and the ground-based observations are used together with ERA5 10 m wind data to inter-
pret the diurnal variation of tropospheric NO$_2$ VCDs.

Maps of tropospheric NO$_2$ VCDs from the GEMS L2 v2.0, the GEMS IUP-UB, and the TROPOMI product, all around the
TROPOMI overpass, show the dominant NO$_2$ hot spot over the SMA and the smaller urban agglomerations. These hot spots,
especially the SMA, show the highest values in the GEMS L2 product. The lowest values are found in the TROPOMI product.
The background NO$_2$ is comparable between the TROPOMI and the GEMS IUP-UB products but is significantly higher in the
GEMS L2 product, presumably because of the different approaches for the stratospheric correction. Due to a missing interpo-
lation of the AMF, the GEMS L2 product shows box structures with the spatial resolution of the GEOS-Chem model.

The evaluation of the three products with the ground-based DOAS measurements shows an overestimation by the GEMS L2
product with a slope of 1.41, a median relative difference of +64 %, and a correlation of 0.75. The evaluation results of the
GEMS IUP-UB and the operational TROPOMI products are comparable. The slope and median relative difference are 0.89
and -1 % for the GEMS IUP-UB product and 0.79 and -14 % for the TROPOMI product. The correlation of the GEMS IUP-UB
improved from 0.82 to 0.85 when observations are limited to the TROPOMI overpass time. This brings the correlation closer
to the 0.88 of the TROPOMI product, indicating larger deviations in coinciding morning and/or afternoon observations.

All comparisons between satellite and ground-based observations are based on the closest pixel within a radius of 5 km around
the station site. Other co-location criteria with different distances, averaging satellite data around the station area and consid-
ering the viewing direction dependency have not significantly improved the comparisons.

The separate comparison of the satellite and ground-based observations for the 11 individual sites illustrates some differences





in agreement between the sites. Correlation for the GEMS IUP-UB product varies between 0.67 for the MAX-DOAS IUP-UB Ulsan site and 0.87 for the MAX-DOAS IUP-UB Incheon site. The slope varies between 0.40 for the MAX-DOAS MPIC Busan site and 1.17 for the Pandora SNU at Seoul National University (SNU). Biases are larger for more polluted sites, while

less polluted sites show differences close to zero. The positive bias for the two least polluted sites is probably related to the stratospheric correction in the GEMS IUP-UB product. In general, the GEMS IUP-UB product and the TROPOMI product show good agreement in the individual biases, supporting the good agreement visible in the overall comparison.

Mobile car DOAS observations serve as an additional data set to evaluate the GEMS observations and support the results obtained from the comparisons with stationary ground-based data.

Due to the locations of the stations in different pollution regimes, the observed diurnal variations of the tropospheric $NO_2$ columns from the GEMS IUP-UB and the ground-based data sets show significantly different characteristics. Urban sites show a maximum of $NO_2$ of varying degrees around 11 local time, while more rural sites show nearly no diurnal variability. For both cases, we find good agreement between the diurnal variability of the GEMS IUP-UB and the ground-based $NO_2$ data. The largest deviations are visible in the morning and especially for the 16:45 KST observation, where the GEMS IUP-UB product

often underestimates the station values.

The separation of the data sets into seasons shows for the polluted sites a minimum in the $NO_2$ columns around noon (13 KST), indicating the larger influence of chemical loss in summer. However, this summer noon minimum is less pronounced in the GEMS observations. Winter observations show, in general, higher $NO_2$ values with rather flat or slightly decreasing $NO_2$ over the day, which is well captured in both data sets. We see no increase over the day, as reported by other studies using total $NO_2$

columns in Seoul and Beijing. Most diurnal variability is found at polluted sites in spring and autumn, with an increase during the morning, a maximum late in the morning or around noon, and a decrease towards the afternoon. Due to the largest data availability, these months dominate the overall diurnal cycle.

Diurnal variability differs significantly for low and high wind speed conditions in both the GEMS IUP-UB and the ground-based data set. However, there is better agreement during windy conditions, likely due to increased dispersion and reduced

inhomogeneities. The influence of dispersion in windy conditions results in observations displaying less diurnal variability. Observations under low wind conditions show strong $NO_2$ increases over the day but only at the most polluted sites, especially during winter. This suggests that, under calm conditions, the reduced dilution and less effective chemical loss in winter are insufficient to offset the accumulating emissions. For a more rural site, the diurnal variability with increasing $NO_2$ values following mean wind patterns for specific seasons and times reveals the impact of transported $NO_2$. Due to a location specific but

for these months characteristic change of wind direction around noon, $NO_2$ pollution of an industrial area is transported close to the station. This is also visible in other areas and on a seasonal basis.

When analyzing the weekday-weekend effect, a good agreement is found between the different products. Depending on the station, the $NO_2$ columns are 20 to 50 % lower on Sundays compared to the weekday average. However, the GEMS L2 product which agrees with the other data sets during weekdays shows significantly less reduction on weekends.

Overall, our analyses revealed significant diurnal variation of $NO_2$. This variation is strongly site-dependent, differs between polluted and less polluted sites, and has location-specific and seasonal characteristics. GEMS IUP-UB and ground-based obser-



vations are in good agreement, which is promising for expanding the analysis of diurnal variation using the extensive GEMS data set. The observed diurnal variation of $NO_2$ offers unique insights into the chemistry and emission of $NO_x$ as well as transport processes, but it needs to be carefully interpreted. These analyses can also help to analyze the upcoming data sets of

the follow-up geostationary air quality missions such as TEMPO over North America and Sentinel-4 over Europe.



*Data availability.* GEMS L2 NO$_2$ data can be accessed at https://nesc.nier.go.kr/en/html/cntnts/91/static/page.do (National Institute of Environmental Research, NIER, 2023). The GEMS IUP-UB NO$_2$ product is available on request. TROPOMI NO$_2$ data are freely available via https://s5phub.copernicus.eu/ (Sentinel-5P Pre-Operations Data Hub, last access: 21 February 2022). The data of Pandora instruments are freely available from the PGN data archive (https://pandonia-global-network.org/, last access: 11 October 2023). The FRM4DOAS MAX-DOAS data are available on request. The ERA5 wind data are freely available from the Copernicus Climate Change (C3S) climate data store (CDS) (Hersbach et al., 2023).

*Author contributions.* All co-authors contributed to the campaign either as participants and instrument operators and/or during campaign preparation. AR provided the GEMS IUP-UB NO$_2$ data product. HL and HH provided information to the GEMS L2 NO$_2$ data product. CF and MMF performed the MAX-DOAS data analysis. The campaign was prepared by HH, LSC, and CKS. KL performed the final data analysis and interpreted the results together with AR. KL wrote the paper with feedback and contributions from all other co-authors.

*Competing interests.* At least one of the (co-)authors is a member of the editorial board of Atmospheric Measurement Techniques.

*Acknowledgements.* We thank the National Institute of Environmental Research of South Korea for providing GEMS data, financial support (NIER-2022-04-02-037), and the excellent organization of the GMAP 2021 and SIJAQ 2022 field campaigns. We thank all participants of the GMAP 2021 and SIJAQ 2022 field campaign. The Deutsches Zentrum für Luft- und Raumfahrt (grant no. 50 EE 2204) is acknowledged for financial support. Copernicus Sentinel-5P level-2 NO$_2$ data are used in this study. Sentinel-5 Precursor is a European Space Agency (ESA) mission on behalf of the European Commission (EC). The TROPOMI payload is a joint development by ESA and the Netherlands Space Office (NSO). The Sentinel-5 Precursor ground-segment development has been funded by the ESA and with national contributions from the Netherlands, Germany, Belgium, and UK. We thank PGN instrument PIs, support staff and funding for establishing and maintaining the Pandora sites used in this investigation. The PGN is a bilateral project supported with funding from NASA and ESA. MMF thanks R. Spurr for the free use of VLIDORT.





**Appendix A**

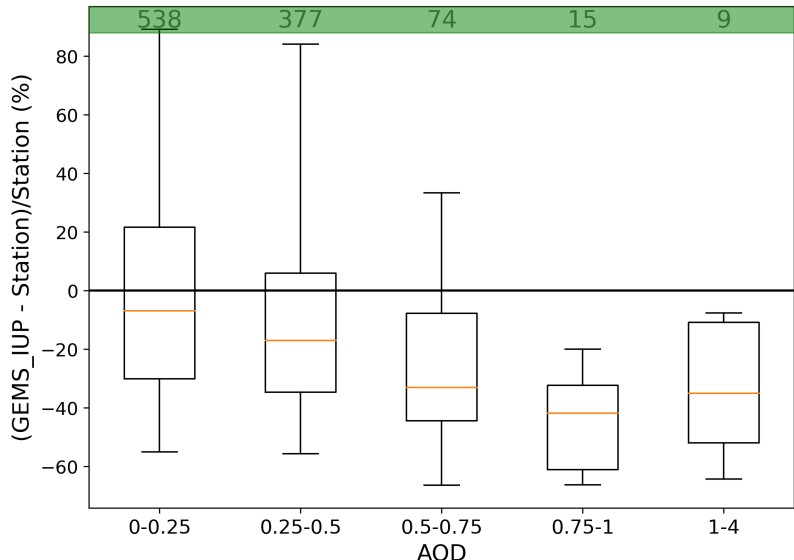

**Figure A1.** Relative median differences between GEMS IUP-UB and MAX-DOAS tropospheric NO$_2$ columns as a function of AOD retrieved within the FRM4DOAS MAX-DOAS NO$_2$ analysis. Numbers in the green bar represent the number of observations contributed to the bin.





**Figure A2.** Scatter plots of GEMS L2 (a), GEMS IUP-UB (b), and TROPOMI NO₂ tropospheric VCDs vs. co-located ground-based NO₂ tropospheric VCDs for different co-location criteria. The time constrain is with ± 20 min the same for all criteria. First row: Ground based measurements within this period are averaged and matched to the closest satellite observation within a radius of 5 km around the station site. Second row: Match to the closest satellite observation within a radius of 10 km. Third row: Satellite pixels are weighted according to their contribution along the line of sight of the ground-based instruments within 5 km of the station. Different VAA are considered independently. Fourth row: Averaging of the VAA comparisons within the ±20 min time window.



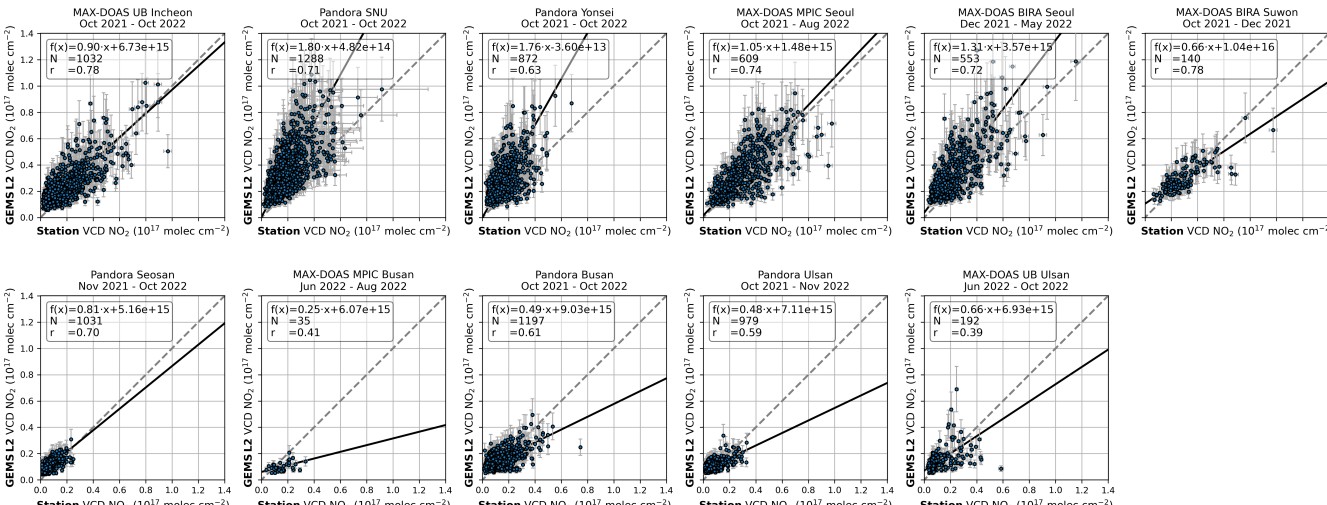

**Figure A3.** Same as Fig. 5 but for GEMS L2 tropospheric NO$_2$ VCDs vs. co-located ground-based NO$_2$ tropospheric VCDs for the 11 individual stations.

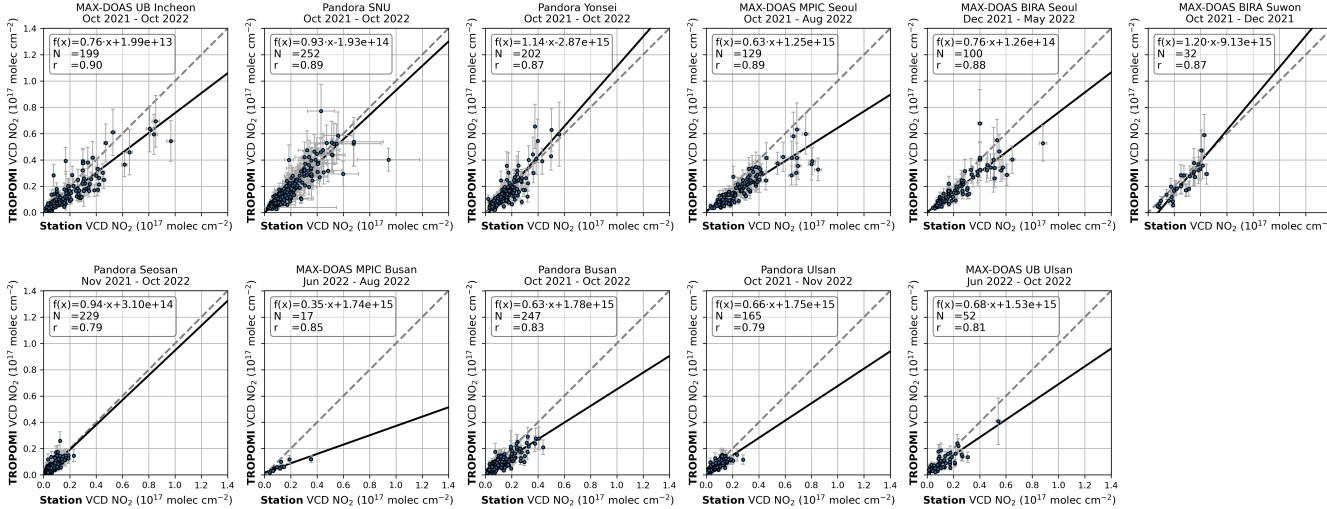

**Figure A4.** Same as Fig. 5 and A3 but for TROPOMI tropospheric NO$_2$ VCDs vs. co-located ground-based NO$_2$ tropospheric VCDs for the 11 individual stations.







**Figure A5.** Same as Fig. 8 but for the GEMS L2 product. Diurnal variability of median tropospheric NO$_2$ VCDs from the GEMS L2 v2.0 product (blue) and ground-based stations (red).



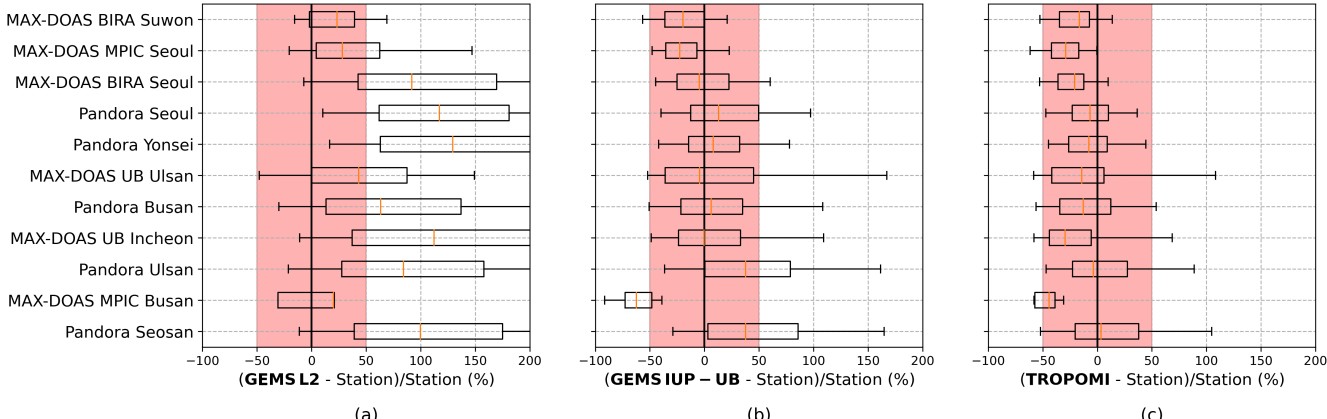

**Figure A6.** Same as Fig. 6 but GEMS L2 and GEMS IUP-UB observations are limited to the TROPOMI overpass time between 12:28 KST and 14:37 KST.





**Figure A7.** Diurnal variability of median tropospheric NO$_2$ VCDs from the GEMS IUP-UB product (blue, light blue) and ground-based stations (red, light red) for weekdays (Monday-Friday), respectively weekends (Saturday and Sunday). Error bars represent the 25 and 75 % quantiles of the MAX-DOAS and GEMS observations. Station names and measurement periods can be found in the individual titles.



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
