# Peer review of "Validation of GEMS tropospheric NO2 columns and their diurnal variation with ground-based DOAS measurements"

_EGUsphere, 2024_

## Author Comment (AC1)

**RC1**: 'Comment on egusphere-2024-617', Anonymous Referee #1

Legend: Referee comments in **blue**, author comments in **black**

Lange et al. (2024) provide thorough validation of the GEMS NO2 column against ground-based DOAS measurements. The work also provides the comparison of diurnal variation as observed by GEMS and ground-based DOAS measurements.

In my opinion, this work deserves publication. My main concern is the readability of the work. It would be beneficial to condense the figures and main text discussions to help the readers take out the key scientific messages from the work. Most of the comments are related to helping the authors to achieve such a goal.

Recommendation: Minor Revision

We would like to thank the reviewer for their helpful comments and for helping to improve the readability by suggestions on how to condense the figures and main text. We hope that we have adequately answered all questions and that our explanations are satisfactory.

General comment about larger changes we have made during the review process:

Upgrade to the new pandora version for pandora Seoul and changes in quality filtering for all pandora data, results in slight changes in several figures involving pandora data and improved data availability for some seasons and sites. For all pandora except the pandora Seoul, the most recent data version (rnvh3p1-8) was available when writing the manuscript. For pandora Seoul, we used the available data product at that time, which was rnvh1p1-7. In the meantime the new version is available for pandora Seoul. Since the column retrieval was improved and changes in the columns are expected, we decided to update to the new version. Additionally, we adapted the quality filter from filtering low quality and unusable data to filtering only unusable data and introduced instead an additional wrms (Normalized rms of fitting residuals weighted with independent uncertainty) filter. This results in a somewhat higher data availability for some seasons. Since ground-based data are only used when quality filtered satellite observations are available, this acts as a further indirect filter. Overall, values in the comparisons have changed only slightly, and the conclusions drawn from the figures remain the same.

To improve readability we moved some plots to the appendix and condensed the discussion about diurnal variability in section 5 (Diurnal variability of GEMS and ground-based tropospheric NO2 VCDs) into the seasonality subsection.

We added a comparison of subversions of the GEMS IUP-UB product using different stratospheric VCD products in the discussion in section 5.4.

**Specific Comments**

Line 95: These two other works seem relevant for references: Oak et al. 2024 (https://doi.org/10.5194/egusphere-2024-393) and Edwards et al. 2024 (https://doi.org/10.5194/egusphere-2024-570).

Thank you for mentioning these two other relevant references. As these papers were published as preprints around the time of our submission, they were not considered in our publication yet but a discussion of their results have been added now.

Lines 165 – 167: Does GEMS not correct for instrument polarization sensitivity and scene inhomogeneity? I'm curious if this correction is unique to GEMS IUP-UB retrieval.

The operational GEMS product currently includes no correction for instrument polarization sensitivity and scene inhomogeneity. We added a comment in the manuscript that this is one of the differences between the two retrievals to make this clear.

Section 2.1.2: What's the rationale behind redoing the DOAS fit for GEMS IUP-UB retrieval? In lines 160 – 162, the most of uncertainties seem to be dominated by AMF calculation. How does it affect the final VCD quality?

Thank you for your comment. We agree that the uncertainty of the tropospheric $NO_2$ VCD is dominated by uncertainties of the AMF calculation and not the DOAS fit. However, redoing the fit provides the opportunity to use a larger fitting window, including polarization correction, destriping, and correction of scene inhomogeneities. All these factors improve the product and result in less noise, a reduction of scatter and improved consistency with other products (TROPOMI, GOME-2) using similar fit windows. The changes improve the product especially over challenging regions. The study area of Korea is much less affected by these changes in the fit. Nevertheless, this study can be used as a first validation of the GEMS IUP-UB $NO_2$ product. This will be extended to a larger region in future studies.

Section 2.1.1: More details about L1 to SCD for GEMS official product would be beneficial just like how great details are provided for GEMS IUP-UB retrieval (lines 164 - 168).

GEMS irradiance data are wavelength calibrated using the pre-launch spectral response function. A single wavelength calibration is applied across all rows. $NO_2$ slant column densities (SCDs) are retrieved from Level 1 spectra using a DOAS fit in the fitting window of 432-450nm. In comparison to the GEMS IUP-UB product, the operational SCD retrieval does not include polarization correction, de-striping, or correction of scene inhomogeneities.

Line 217: "The here used tropospheric $NO_2$ VCDs are" might need a grammar fix.

Changed to: The tropospheric $NO_2$ VCDs used in this study were retrieved by….

Section 2.3: What's the quality difference in Pandora data between the direction sun measurements vs. multi-axis mode? Doesn't multi-axis mode require more assumptions? It might be beneficial to provide the advantages and disadvantages of multi-axis mode.

Thank you for your comment. Yes, direct-sun columns are particularly beneficial for validation/evaluation due to their low uncertainties in the AMF (Herman et al., 2009). However, the Pandora direct sun mode retrievals provide only total vertical columns and not tropospheric $NO_2$ VCDs, which are needed for the validation of tropospheric VCDs. For the conversion of the total into tropospheric $NO_2$ VCDs, additional information on the stratospheric column is needed. Often this is taken from measurements, retrieved from satellite or a climatology. This makes the tropospheric VCD dependent on an input, which is part of the validation process or a climatology, which may not be representable for the measured conditions.

Lines 253 – 254: Perhaps change "hot spots" to a different language. They have elevated $NO_2$ concentration against the background.

Done.

Line 259: Ulsan is not a remote region. The authors mentioned in line 126 that Ulsan is an important industrial center. Similarly, in lines 383 – 384, I wonder if Ulsan low-polluted sites.

You are right, this is misleading. Ulsan, in general, is not a remote region but an important industrial center. However, the Pandora mentioned here is located at the Ulsan National Institute of Science and Technology, located several kilometers outside the city and industry area of Ulsan, which is also visible in Fig. 1 (c). We added a sentence here to clarify that the site location is quite remote but still in the possible area of influence of Ulsan, an important industrial center.

Lines 260 – 265: It might be better to move this to line 249. Otherwise, readers will wonder why GEMS L2 v2.0 has a much coarser resolution than GEMS IUP-UB v1.0 as soon as they see Figure 2. At first, I was wondering how GEMS IUP-UB v1.0 has a finer pixel size than GEMS L2 v2.0 when GEMS IUP-UB v1.0 uses a coarser resolution of chemical transport model for the computation of AMF.

Thanks, we have moved this line forward.

Section 4: It might be better to move Figure 3 to the SI and make the discussion about Figures 3 and 4 more concise. It might be helpful to move 308 – 325 to the Appendix except for lines 321 – 323.

We think Fig. 3 is an essential plot, but moved Fig. 4 (scatter plots restricted to TROPOMI overpass times) to the Appendix and made the discussion more concise. We also moved a large part of lines 308 – 325 about the co-location criteria to the appendix.

Lines 351 – 353: These lines don't add value to the scientific discussion in Section 4.1. Correct me if I am wrong, but it seems obvious that GEMS IUP-UB production and TROPOMI product will show good agreement as they use similar retrieval processes.

We think it is valid to point out that "the GEMS IUP-UB product and the TROPOMI product show good agreement in the individual biases, which is improved when limiting the GEMS IUP-UB product comparisons to the TROPOMI overpass time." The retrieval processes are similar but not identical, and in addition, instruments and observation geometries are different which might cause differences.

Figure 5: It might be beneficial to move this figure to the SI as not all sites are discussed in detail in Section 4.1. Showing only the sites that are discussed but moving the rest to the SI might be helpful to improve the readability.

Thank you for your suggestion. We moved Fig. 5 completely to the appendix, Fig. 6 (Box-and whisker plots) provides more or less the same information in a more concise way.

Section 4.1: I wonder if Section 4 and Section 4.1 can be combined into one and become more concise.

Thank you for this suggestion, we combined both sections.

Line 361 – 362: A correlation of coefficients of 0.87 and 0.82 are similar. No need to mention that 0.87 is better.

We deleted this sentence.

Lines 365 – 358: These lines seem unnecessary.

You mean lines 365 – 368 about the independent car DOAS data retrieval? We moved the lines to the beginning of this section, where they should fit better.

Section 4.2: It would be helpful to add more scientific discussion to this section. What additional information can we obtain from this section that is different from Sections 4 and 4.1? Can they become one section and be more concise? The conclusion seems to be the same in that GEMS IUP-UB underestimates against car DOAS.

Thank you for your suggestion. You are right the comparison of the GEMS IUP-UB to the car DOAS data sets provides only limited additional information. However, it demonstrates that these observations are a valid additional validation data set. Since they can cover a larger part of the GEMS pixel area, they can provide additional insights into the representativeness of observations, which is mentioned and illustrated by the horizontal bars indicating the 10th and 90th percentile of car DOAS observations within the GEMS pixel. Further analysis of this data set is beyond the scope of this paper. Since we have already combined sections 4 and 4.1, we are reluctant to include this section there as well.

Lines 373 – 374: Perhaps change to "GEMS is the first geostationary instrument providing hourly NO2 data. We compare the diurnal variations observed by GEMS and ground-based instruments."

Thank you for the suggestion, we changed the sentence.

Lines 377 – 379: These can go to figure captions.

These lines were deleted during the merging of sections 5 and 5.1.

Lines 379 – 380: Unnecessary line.

These lines were deleted during the merging of sections 5 and 5.1.

Lines 383 – 390: I suggest reducing these lines to one or two sentences. Most sites show no significant bias against Pandora stations while MPIC Seoul and Suwon show more severe bias in the morning. I am noticing that each station has different months and seasons combined. Could it be stemming from averaging different seasons?

These lines were deleted during the merging of sections 5 and 5.1.

Lines 389 – 390: Explanation as to why would be helpful.

Lines were deleted during the merging of sections 5 and 5.1. We mention the larger biases in the morning and late afternoon now in section 5.1 and added the following:

These differences for observations with larger SZA can be explained by a lower sensitivity and more uncertain AMFs for these scenes, which is amplified for larger aerosol loads and low boundary layer heights in combination with a lack of knowledge of the tropospheric aerosol in the AMF calculation for the GEMS IUP-UB product. This is further discussed in Sect. 5.4.

Line 391: I recommend the line to be rephrased to not contain "interesting". This and other parts of the manuscript.

Done

Line 393 – 399: It would be helpful to include Yang et al. (2023b) and Edwards et al. (2024) in the discussion instead of studies that used the LEO instruments to investigate the diurnal variation. Both studies found similar results as in lines 391 – 393. But I wonder how averaging across different seasons would play a role in interpreting Figure 8.

We decided to remove most of section 5 and with it the discussion of Fig. 8 and moved only some key information to the seasonality section. We think it is also interesting to compare to the LEO instrument studies to illustrate the new information GEMS provides. We also included Yang et al. (2023b) and Edwards et al. (2024) in the discussion.

Lines 399 – 401: Crawford et al. 2021 (https://doi.org/10.1525/elementa.2020.00163) and Chong et al. 2019 (10.4209/aaqr.2017.09.0341) might be helpful to interpret this diurnal variation.

Thank you, we were not aware of the study by Chong et al. 2019. The lines you mentioned here were deleted during the merging process. However, we added some more discussion and references in section 5.1.

Section 5: As mentioned in lines 403 – 405, the diurnal variation can vary significantly depending on the season. I would like to suggest moving Figures 8 and 9 to SI and removing Section 5. If there is beneficial information that Section 5 can provide that Section 5.1 cannot, keep both sections.

Thank you for this suggestion. We decided to remove most of section 5 and moved some of the information to the seasonality section. We removed Figure 8 (diurnal variability of station and GEMS IUP-UB data for the individual sites), the related Fig. A5 for GEMS L2 in the appendix, and Fig. 9 completely. Instead, we added plots showing the diurnal variability of the median relative differences similar to the old Fig. 9 but now separately for the individual seasons in the appendix.

Line 410 – 411, lines 440 - 441: Edwards et al. (2024) and Yang et al. (2023b) found consistent results.

Added.

Line 414 – 415: Yang et al. (2023b) found a more pronounced increase in NO2 VCD over the course of the day in the winter when the wind speed was segregated. Figure 5a showed a similar diurnal pattern in the YSU site as this work where the NO2 decreases after noon. When the wind speed is not segregated, the transport term offsets the emission term leading to weaker diurnal variability.

Thank you for your comment. We discuss the different wind conditions in the following section and also compare to Yang et al. there. For the section you are referring to, we clarified that Yang et al. (2023b) only found the increase for the total columns of GEMS, Pandora Beijing and Pandora Seould but not for Pandora Yonsei.

Lines 415 – 420: It seems like the diurnal variation of NO2 in spring is more similar to the summer following what is being described. The autumn seems to show diurnal variation more similar to that of the wintertime based on SNU and Yonsei sites.

The polluted stations in the SMA (MAX-DOAS UB Incheon, Pandora Seoul, Pandora Yonsei) show quite similar diurnal variations with increasing NO2 in the morning, a maximum close to noon around 11/12 KST, and a decrease towards the evening in spring and autumn. We think it is difficult to compare autumn to winter since there the morning and late afternoon observations are missing. Summer observations have a tendency to have a minimum around noon and a slight increase of NO2 in the late afternoon. This is not visible in the spring and autumn data, where the curves are flatter after noon. We added the description of the diurnal variation of the spring and autumn observations to the text and hope it is more coherent and intelligible.

Lines 421 – 432: It may be better to focus on interpreting the diurnal variation in this section.

See answers to previous comments. We focus now on interpreting the diurnal variation in this seasonality section.

Figure 11: This figure is too busy. It might be better to move it to the SI.

We moved it to the appendix.

Line 445: Not sure what "but already when considering all data" means.

This sentence was rephrased.

Figure 13. I wonder if the result for one site can be shown and move the other sites that are not thoroughly discussed into the SI section. This might be helpful for readability.

The figure includes already only four sites. As three out of four are discussed, and to maintain comparability with Fig. 12, which shows the sites for calm conditions, we decided to keep all sites for this figure.

Lines 450 – 451: No need for a separate paragraph for two sentences.

Removed.

Figures 14 and 15. Can it be condensed to one hour in the morning, noon, and afternoon? It might have to make the figure more concise. The main difference seems to be in 7:45 KST, 11:45 KST, and 16:45 KST. Perhaps one of Figures 14 and 15 can be shown in the main text and the other one in the SI as the general conclusion is the same for both sites.

Thank you for your comment. We think that these hourly observations are the big strength of GEMS and that it is important to show that it is possible to observe quite continuous changes with this hourly resolution and better visualize and understand the evolution of NO2. But we

understand that these are busy figures, and therefore, we only kept Fig. 14 for the southeast of South Korea, which is discussed in more detail, and moved Fig. 15 to the Appendix.

Figure 16: Can only a few months be shown that shows a prominent difference?

We tried to reduce the figure to a few months, but it is more difficult to see that the features change almost continuously. Therefore, we decided to keep the figure as it is. Since we moved the maps for the SMA to the Appendix and also reduced the following figure (see next comment), we hope that this part of the manuscript is less busy now.

Figure 17. It might be helpful to just show a few sites on which Section 5.3 discusses and move other figures to the SI.

Thank you for your suggestion. We kept the four sites discussed in the text and moved the other four sites to the appendix.

---

## Author Comment (AC2)

**RC2**: 'Comment on egusphere-2024-617', Anonymous Referee #2

Legend: Referee comments in **blue**, author comments in **black**

The manuscript by Lange et al. presents a very thorough evaluation of the performance of the operational GEMS tropospheric NO2 product and the scientific NO2 product from the University of Bremen over Seoul. The relevant aspects of the retrieval are evaluated in an exhaustive manner: absolute magnitude, seasonality, weekend effect, and diurnal cycle. Also some plausible interpretation of the measurements is provided in terms of emissions, transport, and atmospheric chemistry, which strengthens the study. Overall, I fully support publication of this work, and only make a few remarks and suggestion for corrections below. I agree with the other reviewer that the amount of material presented is quite overwhelming and some condensing would benefit the readability of the paper.

We would like to thank the reviewer for their helpful comments. We hope that we have adequately answered all questions and that our explanations are satisfactory.

General comment about larger changes we have made during the review process:

Upgrade to the new pandora version for pandora Seoul and changes in quality filtering for all pandora data, results in slight changes in several figures involving pandora data and improved data availability for some seasons and sites. For all pandora except the pandora Seoul, the most recent data version (rnvh3p1-8) was available when writing the manuscript. For pandora Seoul, we used the available data product at that time, which was rnvh1p1-7. In the meantime the new version is available for pandora Seoul. Since the column retrieval was improved and changes in the columns are expected, we decided to update to the new version. Additionally, we adapted the quality filter from filtering low quality and unusable data to filtering only unusable data and introduced instead an additional wrms (Normalized rms of fitting residuals weighted with independent uncertainty) filter. This results in a somewhat higher data availability for some seasons. Since ground-based data are only used when quality filtered satellite observations are available, this acts as a further indirect filter. Overall, values in the comparisons have changed only slightly, and the conclusions drawn from the figures remain the same.

To improve readability we moved some plots to the appendix and condensed the discussion about diurnal variability in section 5 (Diurnal variability of GEMS and ground-based tropospheric NO2 VCDs) into the seasonality subsection.

We added a comparison of subversions of the GEMS IUP-UB product using different stratospheric VCD products in the discussion in section 5.4.

General remarks

\* It may be useful to discuss the differences between stratospheric NO2 in the operational, IUP-UB and TROPOMI products in more detail. Especially since stratospheric NO2 is argued to be one of the reasons for the overestimation in the operational GEMS product. Is there a clear reason why the method from Bucsela et al. (2013) would result in too low stratospheric columns?

Thank you for your comment. To investigate the influence of the different stratospheric VCD products, we created subversions of the GEMS IUP-UB product using different stratospheric column products. Figure 1 shows scatter plots of coincident satellite and ground-based

tropospheric NO2 VCD observations for (a) the original GEMS IUP-UB product using the STREAM-based stratospheric VCDs, (b) the GEMS IUP-UB using the TM5 stratospheric VCDs, (c) the GEMS IUP-UB using the GEMS L2 stratospheric VCDs, and (d) the original GEMS L2 product. Replacing the STREAM-based stratospheric VCD with the TM5 data increases the bias from 3% (-22% - 38%) to -20% (-41% - 5%) and changes the offset from +1.6e15 to -3.9e14 molec cm-2. This illustrates that the TM5 model stratospheric VCDs are too large, resulting in too low and even negative tropospheric NO2 VCDs. Using the GEMS L2 stratospheric VCDs for the GEMS IUP-UB product increases the bias and the offset, illustrating that the GEMS L2 stratospheric VCD product is too low. This results in an overestimation of the GEMS IUP-UB tropospheric NO2 VCD compared to the station data. The correlation stays constant for both subversions as there is little correlation between the stratospheric NO2 columns and the tropospheric NO2 variations at the stations. The higher scatter seen in the operational GEMS L2 product is caused by the surface reflectivity as shown in Sect. 5.4.

[Figure]

Figure 1. Scatter plots of satellite vs. co-located ground-based NO₂ tropospheric VCDs; for (a) the original GEMS IUP-UB product using the STREAM-based stratospheric VCDs, (b) the GEMS IUP-UB using the TM5 stratospheric VCDs, (c) the GEMS IUP-UB using the GEMS L2 stratospheric VCDs, and (d) the original GEMS L2 product.

Unfortunately, it is not yet clear why the Bucsela et al. (2013) based GEMS L2 stratospheric VCD is too low, one possible reason might be the chosen threshold value to find tropospheric contamination.

Have the stratospheric NO2 columns been validated?

The stratospheric NO2 columns haven't been validated in this study. Some preliminary evaluation of the GEMS L2 v2 stratospheric product was done within the PEGASOS ESA project. The TROPOMI stratospheric NO2 column was validated by Verhoelst et al. (2021) using zenith-sky DOAS measurements during twilight, showing a slight negative median difference for the stratospheric column data of -2% in summer and -15% in winter. Unfortunately, it is beyond the scope of this paper to go further into details regarding the validation of stratospheric GEMS data.

* How does stratospheric NO2 change throughout the day in the two GEMS products studied?

Thank you for this question. We investigated this in more detail and created plots showing in dark blue the stratospheric NO2 VCD of the GEMS L2 product based on the method from Bucsela et al. (2013), in light blue the stratospheric NO2 VCD based on the STREAM algorithm, and in green the TM5 model stratospheric columns, used in the TROPOMI product which is shown in black. As expected, the TM5 and TROPOMI stratospheric NO2 VCDs agree well. The GEMS IUP-UB STREAM-based stratospheric column shows a very similar diurnal evolution as the TM5 data but is slightly lower. The GEMS L2 product shows a similar but reduced variability over the day and is lower by a factor of around 2.5 when compared to the TM5 and GEMS IUP-UB stratospheric columns. We added plots and discussion in Sect. 5.4 of the manuscript.

[Figure]

Figure 2. Diurnal variability of median stratospheric $NO_2$ VCDs for the GEMS L2 product based on the method from Bucsela et al. (2013) in dark blue, the GEMS IUP-UB STREAM-based product in light blue, and in green the TM5 model stratospheric VCDS, used in the TROPOMI product which is shown in black.

* A clear message what the authors think is the main reason for better validation results around noon than in the morning or late afternoon would be appropriate.

Thank you for your comment. We mention the larger biases in the morning and late afternoon now in section 5.1, and added the following:

"These differences for observations at larger SZA can be explained by a lower sensitivity of GEMS and more uncertain AMFs for these scenes, which is amplified for larger aerosol loads and low boundary layer heights in combination with a lack of knowledge of the tropospheric

aerosol in the AMF calculation for the GEMS IUP-UB product. This is further discussed in Sect. 5.4."

and point to the discussion in section 5.4, where this question is also discussed.

"Another already mentioned aspect, which possibly contributes to the differences, especially at larger SZA, is the lack of knowledge of tropospheric aerosol in the calculation of the AMF for the GEMS IUP-UB product. However, the L2 product considers aerosol parameters from GEMS observations in the AMF determination and should correct for their influence. The expected improvement is not reflected in the comparisons.
Due to less sensitivity at higher SZA (and VAA), AMFs are expected to be more uncertain for these scenes. This uncertainty is further enhanced for larger aerosol loads and with low boundary layer heights in the morning and evening."

We hope that our text is now clearer.

Minor issues

L179-181: does the GEMS IUP-UB product have a similar quality assurance flagging system as TROPOMI?

Yes, the GEMS IUP-UB and TROPOMI quality flagging system are similar but the GEMS IUP-UB has not yet a full error propagation. We have added this in the text.

L291: typo 'sight' --> slight

Changed.

L445-446: this sentence was a bit difficult to follow. Please consider rephrasing.

This sentence was rephrased. We hope it is better to follow now.

L512: stratospheric NO2 columns are usually on the order of 10^15 molec. cm-2

Yes, thank you for the comment, we changed this to 10^15.